# Stable isotopic composition of top consumers in Arctic cryoconite holes: revealing divergent roles in a supraglacial trophic network

Tereza Novotná Jaroměřská[1], Jakub Trubač[2], Krzysztof Zawierucha[3], Lenka Vondrovicová[2], Miloslav Devetter[4,5] and Jakub D. Žárský[1]

[1]Department of Ecology, Faculty of Science, Charles University, Prague, 128 44, Czech Republic
[2]Institute of Geochemistry, Mineralogy and Mineral Resources, Faculty of Science, Charles University, Prague, 128 43, Czech Republic
[3]Department of Animal Taxonomy and Ecology, Adam Mickiewicz University, Poznań, 61-614, Poland
[4]Biology Centre, Institute of Soil Biology, Czech Academy of Sciences, České Budějovice, 370 05, Czech Republic
[5]Centre for Polar Ecology, University of South Bohemia, České Budějovice, 370 05, Czech Republic

*Correspondence to*: Tereza Novotná Jaroměřská (jaromert@natur.cuni.cz)

**Abstract.** Arctic cryoconite holes represent highly biologically active aquatic habitats on the glacier surface characterized by the dynamic nature of their formation and functioning. The most common cryoconite apex consumers are the cosmopolitan invertebrates – tardigrades and rotifers. Several studies have highlighted the potential relevance of tardigrades and rotifers to cryoconite holes' ecosystem functioning. However, due to the dominant occurrence of prokaryotes, these consumers are usually out of the major scope of most studies aiming at biological processes on glaciers. The aim of this descriptive study is to present pioneering data on isotopic composition of tardigrades, rotifers and cryoconite from three High Arctic glaciers in Svalbard and discuss their role in cryoconite hole trophic network. We found that tardigrades have lower $\delta^{15}$N values than rotifers, which indicates different food requirements or different isotopic fractionation of both consumers. The $\delta^{13}$C values revealed differences between consumers and organic matter in cryoconite among glaciers. However, the mechanistic explanation of these variations requires further investigation focused on the particular diet of cryoconite consumers and their isotopic ratio. Our study introduces the first observation of carbon and nitrogen stable isotopic composition of top consumers in cryoconite holes analysed by an improved method for cryoconite sample processing paving the way for further studies of the supraglacial trophic network.

## 1    Introduction

Supraglacial habitat, the environment on the glacier surface, is subjected to a continuous as well as episodic input of allochthonous and autochthonous material and forms a biogeochemical reactor involving a variety of biotic and abiotic processes. Most of the biotic activity is usually connected to ablation zones (areas with an ice loss exceeding its increase) which have a global tendency to increase their surface area due to the climate change (Hodson et al., 2008; Stibal et al., 2012a). Moreover, the export of the biological communities and their metabolic production has a potential to influence the downstream deglaciated areas and coastal marine ecosystems (Bardgett et al., 2007; Foreman et al., 2007; Hodson et al., 2008; Hood et al., 2009; Williams and Ferrigno, 2012).

The accumulated fine material on the glacier surface (so called cryoconite, Nordenskiöld (1875)) – due to its dark colour – reduces albedo of the glacier ice and creates water-filled depressions called cryoconite holes (Cook et al., 2016; Takeuchi et al., 2001). The diameter and the depth of cryoconite holes usually range from a few centimetres to tens of centimetres (Gerdel and Drouet, 1960; Fountain et al., 2004; Zawierucha et al., 2018a; Zawierucha et al., 2019a). At the bottom of the holes, cryoconite forms aggregates (cryoconite granules) composed of bacteria,

organic and inorganic matter (Takeuchi et al., 2001) which provide a suitable environment for various organisms (Zawierucha et al., 2015; Zawierucha et al., 2021). The supply of organic and inorganic matter into cryoconite holes is restricted to allochthonous input from atmospheric deposition, weathering of mineral dust, aeolian deposition, and locally from bird guano deposition (Anesio et al., 2009; Benassai et al., 2005; Edwards et al., 2014; Hodson et al., 2005; Stibal et al., 2008; Telling et al., 2011; Vonnahme et al., 2016; Xu et al., 2010; Žárský et al., 2013). The autochthonous input of matter is generally restricted to microbial activity and recycling (Telling et al., 2011; Telling et al., 2012). Moreover, adjacent areas of glaciers can vary a lot in terms of topography, geology, vegetation and stage of soil development. Therefore, the allochthonous matter brought to the glacial surface can influence the composition of its surface material and biota (Grzesiak et al., 2015; Marshall and Chalmers, 1997; Stibal et al., 2008).

Cryoconite holes cover about 7 % of the surface of the ablation zone (Bøggild et al., 2010; Fountain et al., 2004; Stibal et al., 2012b) and form the most nutrient-rich and biologically active habitats within the supraglacial environment (Cameron et al., 2012; Hodson et al., 2008). As mentioned by Säwström et al. (2002), the rate of photosynthesis in cryoconite holes is comparable with rates of arctic polar lakes and consequently the rate of respiration and utilization of organic matter is very high (Hodson et al., 2008). Thus, cryoconite holes form an important net carbon sink or source in polar ecosystems which depends on the balance between autotrophic and heterotrophic production (Stibal et al., 2012a). Moreover, due to their high biological activity, cryoconite holes efficiently retain nutrients (Bagshaw et al., 2013) and the accumulated matter can consequently provide a source of important nutrients into adjacent areas (Anesio et al., 2010; Porazinska et al., 2004). Therefore, the impact of cryoconite holes on glacier ecosystems nutrient pathways (e.g. carbon, nitrogen, and other microelements) and on downstream ecosystems is a key component for an understanding of the glacial ecosystems functioning (Anesio et al., 2010; Bagshaw et al., 2013; Stibal et al., 2012a; Telling et al., 2011).

Organisms inhabiting cryoconite holes range from bacteria, algae and fungi to metazoans such as tardigrades (phylum Tardigrada) and rotifers (phylum Rotifera) (Cook et al., 2016; Kaczmarek et al., 2016; Zawierucha et al., 2015; Zawierucha et al., 2021). Tardigrades and rotifers are cosmopolitan microscopic invertebrates contributing to multiple aquatic and terrestrial trophic levels as carnivorous, herbivorous, omnivorous and microbivorous species (Guidetti et al., 2012; Guil and Sanchez-Moreno, 2013; Hallas and Yeates, 1972; Kutikova, 2003). Due to their ability to survive various extreme conditions (Guidetti et al., 2011; Ricci, 2001), these animals represent a large component of microfauna in polar and high mountain regions and are the exclusive metazoans inhabiting cryoconite holes in the Arctic (Zawierucha et al., 2018a; Zawierucha et al., 2019b).

As the top consumers of Arctic cryoconite holes, tardigrades and rotifers may represent an important driver of the community of primary producers by grazing and nutrient recycling, thus setting stoichiometric constraints on the local community (Elser and Urabe, 1999; Vonnahme et al., 2016; Zawierucha et al., 2018a; Zawierucha et al., 2021). Previous research on biota from cryoconite holes on Svalbard archipelago revealed that the size distribution and concentration of algae, particularly Zygnematales and Chlorococcales, correlates with the community structure of consumers represented by tardigrades and rotifers (Vonnahme et al., 2016). Documented correlations indicate that grazing likely has an impact on the structure of primary producers in cryoconite holes and presumably contributes to available nutrient quantities and ratios in cryoconite. Nevertheless, other studies from the margin of the Greenland ice sheet revealed a lack of quantitative relations between the numbers of top consumers and

potential food such as cyanobacteria and algae (Zawierucha et al., 2018a) and demonstrated the variability of supraglacial systems which is influenced by multiple factors occurring on various glaciers (Porazinska et al., 2004). As described by Střítecká and Devetter (2015), tardigrades and rotifers are efficient filtrators and especially rotifers reveal high filtration rates in cryoconite holes. The feeding behaviour and morphology of the feeding apparatus indicate that cryoconite species consume mostly algae, bacteria and detritus (Devetter, 2009; Iakovenko et al.,

2015; Zawierucha et al., 2016). However, their diet in various environments differs interspecifically (Guidetti et al., 2012; Guil and Sanchez-Moreno, 2013; Hallas and Yeates, 1972; Kutikova, 2003; Mialet et al., 2013; Wallace and Snell, 2010; Zawierucha et al., 2016).

Analyses of stable isotopes are a well-developed tool which enables us to uncover the trophic interactions of organisms within various systems (McCutchan et al., 2003; O'Reilly et al., 2003; Wada, 2009; Yoshii et al., 1999).

Because of the differences in isotopic fractionation, $\delta^{13}C$ and $\delta^{15}N$ isotopic values of organisms and their potential food can reflect their possible mutual relationships and positions within the food web (Michener and Lajtha, 2008). Isotopic fractionation is caused by physical or biochemical processes which favour lighter or discriminate heavier isotopes (Michener and Lajtha, 2008). The $\delta^{13}C$ value reflects the diet of the organism and is similar or slightly higher within the animal compared to its food (Peterson and Fry, 1987). The slight increase between organismal

$\delta^{13}C$ and the $\delta^{13}C$ values of its diet is caused by a higher assimilation of heavier $^{13}C$ supported by the discrimination against $^{13}C$ during respiration (Blair et al., 1985; DeNiro and Epstein, 1978; Ekblad and Högberg, 2000; Wada, 2009). Therefore, the process of consumption and growth generally tends to increase the $\delta^{13}C$ within the consumer's body compared to its diet. However, larger variations in $\delta^{13}C$ are balanced by a higher release of $^{13}C$ during excretion (DeNiro and Epstein, 1978). The $\delta^{15}N$ values reflect the nitrogen isotopic composition of the

organism's diet and point to the position of organisms in a food chain (DeNiro and Epstein, 1981). The $\delta^{15}N$ value is usually higher in the animal body compared to its diet and increases with the trophic level (DeNiro and Epstein, 1981; Kling et al., 1992; Zah et al., 2001). This increase is mostly caused by a higher proportion of proteins within the diet and subsequent preferential excretion of $^{14}N$ during protein metabolism (Kling et al., 1992; McCutchan et al., 2003). Furthermore, if the environment is limited by a specific nutrient, the consumer's body fractionates

isotopes differently than in the case of no nutrient limitation (Michener and Lajtha, 2008; Šantrůček et al., 2018). For example, Adams and Sterner (2000) described that if the diet had a high C:N, the $\delta^{15}N$ of consumers' body increased. Another study demonstrated, that if the diet is limited by a nutrient, the consumers' body tends to increase or decrease the fractionation against heavier isotope to keep its isotopic values almost constant (Aberle and Malzahn, 2007). Stable isotopes of carbon and nitrogen are the most common food web tracers used in

ecological studies (Michener and Lajtha, 2008). In the case of invertebrates, many studies focus on aquatic or soil food webs where producers and consumers can be easily collected and prepared, and their body size enables us to create the required number of analyses with a sufficient number of individuals (e.g. Ponsard and Arditi, 2000; Wada, 2009). Several studies have also focused on carbon and nitrogen stable isotopes in polar areas (Almela et al., 2019; Shaw et al., 2018; Velázquez et al., 2017). However, none of them on glaciers, which are an essential

part of polar ecosystems and high mountain areas.

The primary producers such as cyanobacteria and algae are an important biotic component reflecting differences in the nutrient input on the glacier surface and contributing to the glacial ecosystem functioning (Hodson et al., 2008; Stibal et al., 2012b; Vonnahme et al., 2016). Studies focusing on the role of top consumers in cryoconite

holes are lacking, however, which may hinder our understanding of cryoconite holes' and glacial ecosystems'
ecology. This study is based on data from three High Arctic inland glaciers, all three located in a different
geomorphological and geological context. We expected that different geomorphological characteristics will be
reflected in the input of organic matter and thus in the composition of their consumers (Cameron et al., 2012;
Edwards et al., 2013a; Edwards et al., 2013b). The current state of knowledge about abundances and feeding rates
of glacier invertebrates suggests that they possess a substantial capacity to influence the biotic fluxes of nutrients
and energy on the glacier surface. Therefore, we assume that the activity of invertebrates is likely an important
component of the nutrient recycling in the glacier system with possible implications to downstream ecological
processes. Here we apply the stable isotope analysis to examine whether the top consumers – tardigrades and
rotifers – show probable differences in their food sources in the glacial ecosystem and discuss their trophic position
in cryoconite holes. This work brings the first evidence of differences in food sources or isotopic fractionation in
two widespread groups of glacier invertebrates and poses an important step for the ongoing research focusing on
underlying mechanisms in the observed patterns.

## 2    Material and Methods

### 2.1    Study site and sampling

Samples of cryoconite were collected from three glaciers (Ebbabreen, Nordenskiöldbreen and Svenbreen; *breen*
means glacier in Norwegian) located at Central Svalbard (78° N and 14–17° E) during July and August 2016.
Svenbreen is a representative of small glaciers in the geologically older part of the Billefjorden Fault Zone.
Ebbabreen and Nordenskiöldbreen are larger valley glaciers within a geologically younger zone. On each glacier,
representative cryoconite holes (varied in shape, size and depth) were sampled in the upper (close to the
equilibrium line) and the lower part (closer to the glacier terminus) of the ablation zone around the main axis of
the glacier. Sampling was conducted twice from each glacier (within the interval of approximately one week
between each sampling) using a high-density polyethylene (HDPE) bottle with two siphons according to Mueller
et al. (2001) with modifications after Vonnahme et al. (2016). Sampled cryoconite from each part of the ablation
zone was poured together and put into sterile Whirl-Pak® (Nasco, Fort Atkinson, WI). Water pH was measured
during the sampling by a Hanna Instrument (HI 98130). Data about the air temperature were provided by the
meteorological station at Bertilbreen which is a glacier adjacent to the examined Svenbreen. After sampling,
cryoconite was stored on ice in a field refrigerator (a plastic barrel entrenched into permafrost) and subsequently
frozen at −20 °C and kept frozen until analysis.

### 2.2    Preparation of samples for isotopic analyses

For each replicate, a part of cryoconite (~ 2–4 cm$^3$) was separately melted by dropping distilled water through the
sample into a glass beaker, transferred into a falcon tube and stored in a cooling box. Animals were collected under
a light microscope (Olympus CX31 and Leica DM750) using a glass Pasteur pipette. All work was performed in
nitrile gloves to avoid carbon contamination. Every individual specimen was cleaned from alien particles and
transferred at least once to a drop of clean distilled water before transferring into an Eppendorf tube. The Eppendorf
tubes were also continuously cooled by a cooling pad. The collected individuals were stored in a freezer at −20 °C
until lyophilization and further processing started. After at least 300 individuals of both taxa (tardigrades and

rotifers) were collected from each sample, the Eppendorf tubes were thawed and all individuals from each sample were transferred into a pre-weighted tin capsule (Costech 41077, 5 × 9 mm). If the water content in the capsule exceeded ½ of the volume, capsules were dried inside a desiccator with silica gel (0.5–2.5 h) until the water inside the capsules was reduced to 1/3 of the volume. The samples were consequently frozen at –20 °C and at least half an hour before the lyophilization stored at −80 °C. The duration of the lyophilization was 4 hours. Thereafter, samples were weighed (Mettler Toledo Excellence Plus XP6, linearity = 0.0004 mg), the capsules were closed and wrapped, and analysed immediately or stored in a desiccator until the analyses were performed. The average dry weight of invertebrates in the capsule was ∼ 29.5 μg. Also, since the identification of species requires specific preparation (see the section 2.5), samples for isotopic analyses were pooled samples of all species occurring in used cryoconite. Four replicates of tardigrades, rotifers and cryoconite from Svenbreen, five replicates of tardigrades, four replicates of rotifers and three replicates of cryoconite from Nordenskiöldbreen, and three replicates of tardigrades, two replicates of rotifers and two replicates of cryoconite from Ebbabreen were collected for the isotopic analyses. Due to the adaptation of cryoconite consumers to specific conditions occurring on the glacier surface (e.g. low temperature, low content of available nutrients), we modified commonly used methods to avoid alteration of their chemical composition during the preparation for isotopic analyses. Therefore, we chose lyophilization instead of oven drying because we wanted to avoid any added component which could potentially contaminate samples.

Cryoconite intended for the isotopic analyses was cleaned from tardigrades and rotifers, which were collected in parallel for isotopic analyses described above. After the collection, cryoconite was stored in Eppendorf tubes at −20 °C. When all samples were prepared, cryoconite was homogenised using an agate pestle and mortar and dried in a thin layer on a Petri dish at 45 °C. The duration of drying was 8 hours.

For the analyses of $\delta^{15}N$ in organic matter (OM), cryoconite was transferred without any other preparation into pre-weighed tin capsules (Costech 41077, 5 × 9 mm) and weighed. The average amount of cryoconite used for analyses was ∼ 31 mg. For the analyses of $\delta^{13}C$ in organic matter, 11–12 mg of cryoconite was transferred into pre-weighed silver capsules (Elemental Analyses, 8 × 5 mm, D2008) and carbonates (e.g. calcite, dolomite) were dissolved using 10% HCl moistened with diH$_2$O. The acid was pipetted into the capsules followed by additions of 10, 20, 30, 50 and 100 μL with drying after each addition according to Brodie et al. (2011) with the modification after Vindušková et al. (2019). After the last acid addition, samples were left drying at 50 °C for 17 hours. After drying, silver capsules were inserted into tin capsules and put into a desiccator for 10–20 days.

## 2.3 Stable isotopes analyses

The $\delta^{13}C$ and $\delta^{15}N$ values in all samples were analysed using a Flash 2000 elemental analyser (ThermoFisher Scientific, Bremen, Germany). Released gasses (NO$_x$, CO$_2$) separated in a GC column were transferred to an isotope-ratio mass spectrometer Delta V Advantage (ThermoFisher Scientific, Bremen, Germany) through a capillary by Continuous Flow IV system (ThermoFisher Scientific, Bremen, Germany). The stable isotope results are expressed in standard delta notation (δ) with samples measured relative to Pee Dee Belemnite for carbon isotopes and atmospheric N$_2$ for nitrogen isotopes and normalized to a regression curve based on international standards IAEA-CH-6, IAEA-CH-3, IAEA 600 (International Atomic Energy Agency, Vienna) for carbon and IAEA-N-2, IAEA-N-1, IAEA-NO-3 (International Atomic Energy Agency, Vienna) for nitrogen. The regression

curve of the total gas for analyses of cryoconite was based on the international standard ST-Soil Standard (Peaty, Elemental Microanalysis, UK) and ST-Soil Standard (SSclay, Elemental Microanalysis, UK). Analytical precision as a long reproducibility for standards was within ±0.03 ‰ for $\delta^{13}C$ and ±0.02 ‰ for $\delta^{15}N$.

The isotopic values of nitrogen in OM as well as organic carbon (decarbonized cryoconite) in cryoconite were used as a reference to the isotopic composition of potential food source for the invertebrates.

## 2.4    X-Ray Diffraction

To reveal the differences in geological composition of sediment among the three glaciers, mineral phases of homogenized sediment were determined by an X-Ray diffraction analysis on the PANalytical X'PertPro (PW3040/60) with an X'Celerator detector. The measurements were conducted under the following conditions: radiation – CuKα, 40 kV, 30 mA, angular range –3–70° 2θ, step 0.02°/150 s. The results were evaluated using a X'Pert HighScore Plus software 1.0d program with a JCPDS PDF-2 (ICDD, 2002) database.

## 2.5    Cryoconite holes community composition

For the species identification, at least 10 cm³ of cryoconite was used from each sample. Tardigrades were collected using a glass Pasteur pipette and the first observation was made under a stereomicroscope (Olympus SZ 51). Immediately after collecting, clean tardigrades were transferred onto glass slides and mounted in a small drop of the Hoyer's medium (Anderson, 1954; Ramazzotti and Maucci, 1983). After one day of drying in 56 °C, tardigrades were identified under a light microscope with phase contrast (Olympus BX53) associated with a digital camera ARTCAM 500. Due to the ambiguities associated with the identification of cryoconite species (species complexes and hidden molecular lines (Zawierucha et al., 2020)), tardigrades were classified to the trophic groups based on the dominant feeding behaviour and feeding apparatus morphology according to Guidetti et al. (2012), Guil and Sanchez-Moreno (2013), Hallas and Yeates (1972) and Kosztyła et al. (2016). Specimens of bdelloid rotifers were identified using a compound light microscope when moving (identification is performed using the morphology of their cirri and trophi). Identification of feeding behaviour of rotifers was primarily conducted following the monograph by Doner (1965). For the identification of eukaryotic primary producers, small drops of thawed and well-mixed cryoconite were placed on the mount. Afterwards, algae and cyanobacteria were identified using a light microscope Olympus BX51 equipped with Nomarski interference contrast and the digital camera Olympus EOS 700D. Identification was based on publications by Ettl and Gärtner (2014), Starmach (1966) and Wehr et al. (2015). Quantification of primary producers was omitted due to the preservation of samples by freezing which presumably has a taxon-specific effect on the survival of cells of the phototrophs. This presumption is based on the observed low survival rate of glacial algal cells (*Mesotaenium*, *Ancylonema*) in freeze-thaw cycles (Jakub D. Žárský, personal communication, 2020). The proportional representation of consumers in each sample was calculated during the collecting of tardigrades and rotifers for isotopic analyses and it is presented as frequency (in %) towards the total amount of collected animals on each glacier. A difference in relative abundance lower than 5 % was considered an equal proportion.

## 2.6    Statistical Analyses

All statistical analyses were conducted in R version 3.5.3 (R Development Core Team, 2018). To test the differences between $\delta^{15}N$ isotopic values of tardigrades and rotifers, a Kruskal–Wallis rank sum test was used.

Before the correlation coefficient tests were applied, a Shapiro–Wilk test was used to test the normal distribution of the data. Therefore, the Pearson's rank correlation coefficient was calculated for all the correlations between isotopic values ($\delta^{13}$C and $\delta^{15}$N) of cryoconite and isotopic values of tardigrades and rotifers which were normally distributed. The $\delta^{15}$N of tardigrades and frequency of omnivorous tardigrades among glaciers were non-normally distributed thus the Spearman's product-moment correlation coefficient was used. Correlation coefficients using Shannon–Wiener Index of Diversity were used to reveal differences between species composition and isotopic values ($\delta^{15}$N, $\delta^{13}$C) of tardigrades. To compare isotopic values of tardigrades, rotifers and cryoconite from each sampling site, One-Way ANOVA and Tukey multiple comparisons of means were applied. For the purpose of statistical analyses, all replicates from the same sampling campaigns were averaged.

## 3    Results

### 3.1    Mineral composition and characteristics of cryoconite

X-Ray diffraction of cryoconite showed that the glaciers differ in mineral composition. Svenbreen has a low amount of dolomite and amphibole which are dominantly found within the metamorphic basement rocks around Ebbabreen and Nordenskiöldbreen. The distribution of minerals within each glacier is shown in Table A1 (Appendices). The ANOVA analysis applied on the mean $\delta^{13}$C values of OC in cryoconite did not reveal any significant difference between glaciers. Due to logistical issues, pH in cryoconite holes was measured only on Svenbreen and Nordenskiöldbreen with values pH < 7.

### 3.2    Isotopic values

The isotopic value of nitrogen showed significant differences in $\delta^{15}$N between tardigrades and rotifers in all samples (Kruskal–Wallis chi-squared = 12.685, *df* = 1, n = 22, *p*-value = 0.00037). All measured $\delta^{15}$N values of tardigrades revealed lower $\delta^{15}$N values than rotifers as shown in Fig. 1 and Table 1.

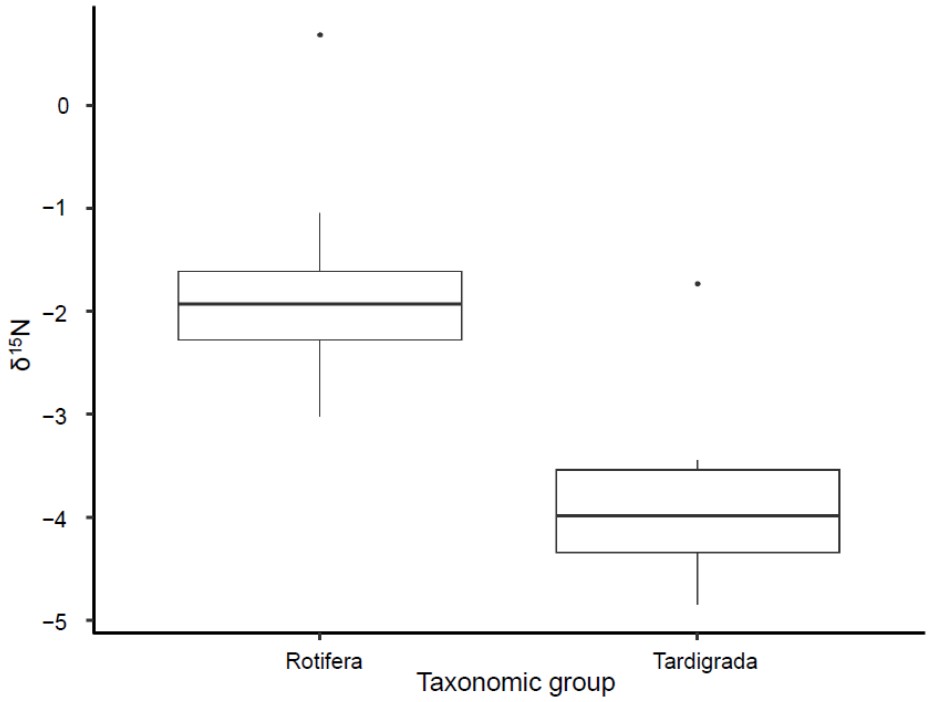

**Figure 1.** Differences in $\delta^{15}N$ between tardigrades and rotifers analysed by Kruskal–Wallis rank sum test. The diagram displays medians and distribution of measured $\delta^{15}N$ values. The whiskers represent the lowest and highest measured values. Both outliers represent $\delta^{15}N$ values of one replicate from Ebbabreen.

**Table 1.** Description of samples and isotopic values ($\delta^{13}C$, $\delta^{15}N$) of tardigrades, rotifers and cryoconite. Isotopic values are presented as ‰ and related to the international standards Pee Dee Belemnite for carbon and atmospheric $N_2$ for nitrogen. The $\delta^{13}C^*$ are values of cryoconite after carbonate removal. The frequency of consumers on each glacier is expressed as % relative to the total amount of collected consumers for isotopic analyses. T signifies tardigrades and R signifies rotifers.

| Glacier | Tardigrades | | Rotifers | | Cryoconite | | Frequency | |
|---|---|---|---|---|---|---|---|---|
| | $\delta^{15}N$ | $\delta^{13}C$ | $\delta^{15}N$ | $\delta^{13}C$ | $\delta^{15}N$ | $\delta^{13}C^*$ | T | R |
| Sven | −3.55 | −23.75 | −1.57 | −27.09 | −2.20 | −22.65 | 39 | 61 |
| | −4.04 | −26.36 | −1.81 | −26.69 | −2.17 | −22.08 | | |
| | −3.52 | −26.33 | −2.04 | −31.16 | −2.24 | −24.09 | | |
| | −4.39 | −26.45 | −1.04 | −28.53 | −1.64 | −23.25 | | |
| Nordenskiöld | −4.39 | −22.91 | −2.20 | −23.36 | −3.58 | −21.56 | 49 | 51 |
| | −3.45 | −23.30 | −1.72 | −23.25 | −2.30 | −20.56 | | |
| | −3.76 | −22.58 | −2.30 | −23.02 | −2.98 | −22.52 | | |
| | −4.15 | −24.78 | −3.02 | −25.19 | | | | |
| | −3.93 | −25.11 | | | | | | |
| Ebba | −1.73 | −24.19 | 0.69 | −26.32 | −2.30 | −22.24 | 58 | 42 |
| | −4.85 | −25.22 | −2.38 | −25.13 | −4.29 | −23.31 | | |
| | −4.33 | −25.15 | | | | | | |


Furthermore, we measured $\delta^{15}N$ values of nitrogen in organic matter from cryoconite, but there was no significant relation with $\delta^{15}N$ values of tardigrades and rotifers found.

Regarding the isotopic values of carbon, we found a positive correlation of the $\delta^{13}C$ values of decarbonized cryoconite and the $\delta^{13}C$ of rotifers (Pearson's product-moment correlation; r = 0.83, n = 19, *p*-value = 0.006) (Fig. 265  2a). The respective relationship among tardigrades was not significant (Pearson's product-moment correlation; r = 0.67, n = 21, *p*-value = 0.07) (Fig. 2b).

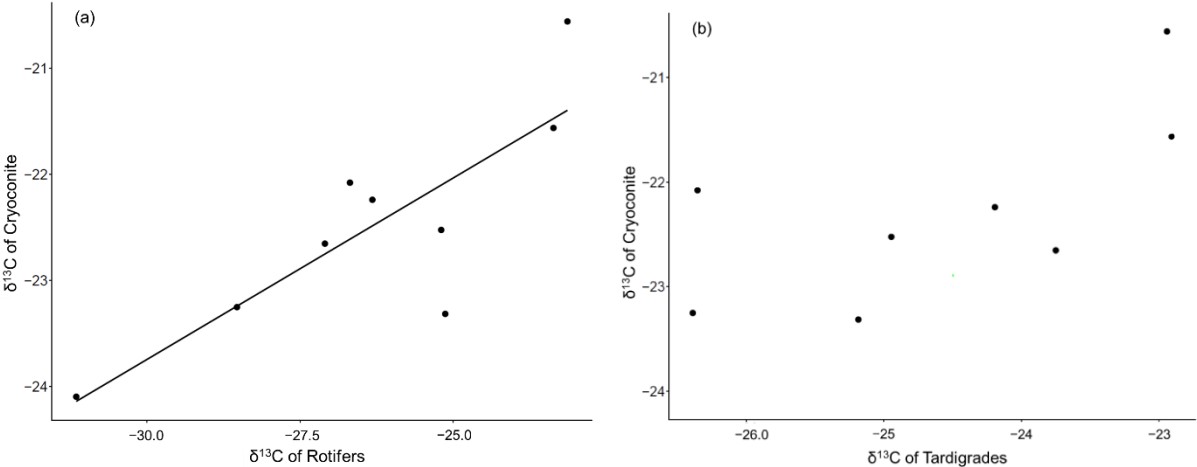

**Figure 2.** Correlation between $\delta^{13}C$ of rotifers with the linear regression line **(a)** and tardigrades **(b)** and decarbonized cryoconite.

In all samples, differences in δ¹³C and δ¹⁵N of tardigrades and rotifers and differences in δ¹³C of decarbonized cryoconite among glaciers were tested using ANOVA test with the mean values of δ¹³C and δ¹⁵N and the Tukey multiple comparisons of means. These analyses showed a significant difference in δ¹³C values of rotifers between glaciers ($p$-value = 0.029) (Fig. 3a), mostly between Nordenskiöldbreen and Svenbreen ($p$-value = 0.025). All other tests did not reveal any significant pattern (Fig. 3).

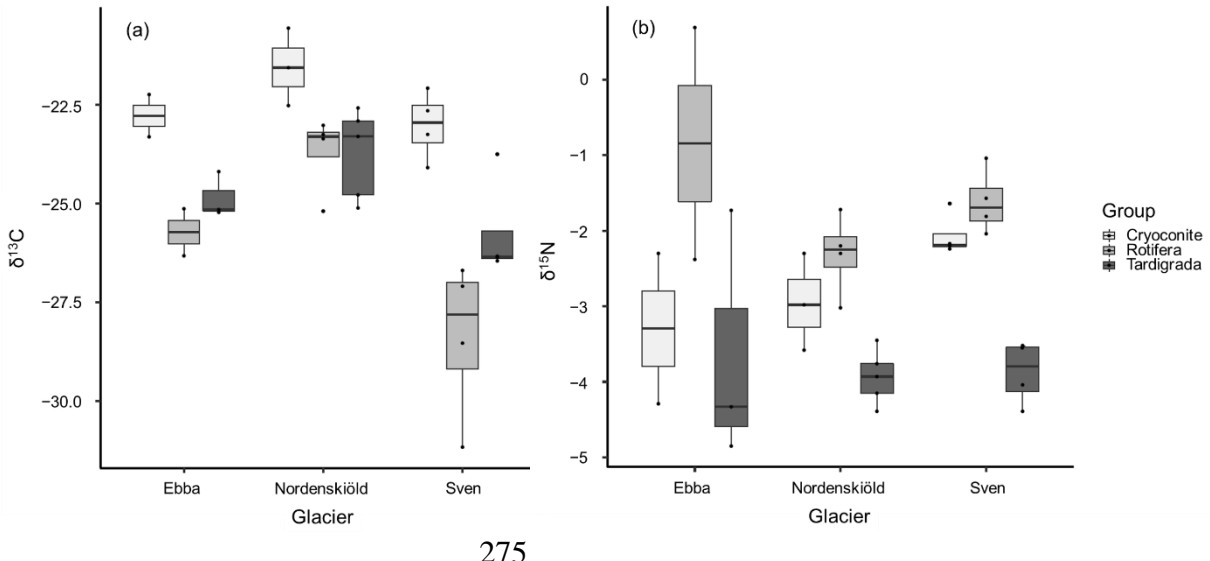


**Figure 3.** Distribution of δ¹³C **(a)** and δ¹⁵N **(b)** isotopic values in tardigrades (n = 12), rotifers (n = 10) and cryoconite (n = 9) among glaciers. The differences in variances among glaciers are the result of the low number of replicates.

### 3.3    Cryoconite holes community composition

During the collection of animals for isotopic analyses, we counted the frequency of tardigrades and rotifers within all replicates (Table 1). On all three glaciers, tardigrades and rotifers co-occurred. However, Svenbreen revealed dominance of rotifers from the total number of 7375 collected individuals, Ebbabreen was dominated by tardigrades from the total number of 5163 collected individuals and Nordenskiöldbreen revealed equal proportion of tardigrades and rotifers from the total number of 6401 collected individuals.

Regarding the species composition of primary producers, we identified representatives of algae and cyanobacteria from all samples. In case of algae, we observed mostly Zygnematales (*Ancylonema* sp., *Mesotaenium* sp.). In case of cyanobacteria, we observed Oscillatoriales (*Phormidium* sp.), Nostocales (*Nostoc*) and Synechococcales (*Leptolyngbya* sp.).

During the division of consumers into trophic groups, only tardigrades were identified in a sufficient number for
analyses. Rotifers found within the samples were identified as *Macrotrachella* sp. and *Adineta* sp. However, they could not be divided and analysed due to the majority of individuals occurring in a dormant stage, which made it impossible to observe the morphology of their cirri and trophi (jaws) necessary for their identification. Regarding tardigrades, we identified 1117 individuals which were divided into three trophic groups: *Pilatobius glacialis* Zawierucha et al., 2020 as microbivores (41 %), hypsibids (*Acutuncus mariae* Zawierucha et al., 2020 and
representatives of *Hypsibius dujardini* group) as herbivores (53 %) and *Grevenius cryophilus* Zawierucha et al.,

2020 as omnivores (5 %). We also found a few individuals of *Cryoconicus kaczmareki* Zawierucha et al., 2018b on Ebbabreen but they were not included into statistics due to their very rare occurrence. As shown in Fig. 4, the composition of tardigrade trophic groups is slightly different among glaciers.

Correlations between trophic groups of tardigrades and isotopic values ($\delta^{15}$N, $\delta^{13}$C) of tardigrades and decarbonized cryoconite did not reveal any significant relationship.

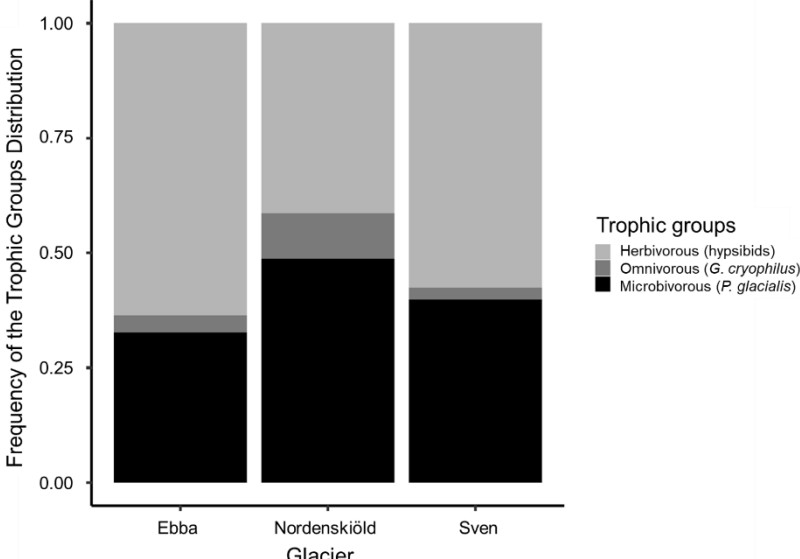

**Figure 4.** Bar plot visualization of tardigrades' trophic groups relative distribution among glaciers.

## 4    Discussion

### 4.1    Isotopic values and the role of consumers in cryoconite trophic network

The nitrogen isotopic values of cryoconite consumers revealed higher $\delta^{15}$N in rotifers compared to tardigrades suggesting differences in $\delta^{15}$N of their diet or differences in the isotopic fractionation between both consumers. Predominantly, higher values of $\delta^{15}$N usually indicate a higher trophic level (Kling et al., 1992; Wada, 2009). However, based on the observed food preferences of tardigrades and rotifers from cryoconite (Střítecká and Devetter, 2015; Zawierucha et al., 2016), we cannot presume their strict trophic division, nor clearly distinguish their feeding strategies. Rotifers were mostly identified as filter feeders (*Macrotrachella* sp.) or scrapers (*Adineta* sp.) (Herzig et al., 2006) whereas *Adineta* sp. did not exceed *Macrotrachella* sp. in the total amount of collected rotifers (the living specimens can be well distinguished from *Macrotrachella* sp. during collecting). Tardigrades found in samples were identified as microbivorous (*P. glacialis*), herbivorous (hypsibids) and omnivorous (*G. cryophilus*) species. Nevertheless, based on the knowledge of the tardigrades' feeding apparatus morphology, *P. glacialis* with its ventrally located mouth is likely able to consume algae during scratching biofilms from the surface of granules, and *G. cryophilus*, which have a relatively wide buccal tube, can utilise various food sources such as algae, protozoans and other small invertebrates. Moreover, all studied groups of tardigrades feed on algae in laboratory cultures (Bryndová et al., 2020; Kosztyła et al., 2016).

Therefore, we assume that the differences in $\delta^{15}$N between both consumers may suggest enrichment of $^{15}$N in food for rotifers caused by preferential consumption of bacteria or DOM and consequent higher $\delta^{15}$N compared

to food for tardigrades (Altabet and Small, 1990; Kling et al., 1992; Mariotti et al., 1980; McCutchan et al., 2003; Peterson and Fry, 1987). The results of Nagarkar et al. (2004) and Kohler et al. (2018) who reported that cyanobacteria have high content of proteins and high $\delta^{15}N$ values typical for nitrogen fixing organisms serve as an indirect empirical indication supporting this assumption. However, lower $\delta^{15}N$ of tardigrades may suggests the variation in $\delta^{15}N$ of algae which can vary depending on their C:N ratio and $\delta^{15}N$ ratio of their nitrogen source

(Adams and Sterner, 2000; Gu and Alexander, 1993). The differences between consumers may also signify different nitrogen isotopic fractionation depending on the C:N ratio of their food (Aberle and Malzahn, 2007; Adams and Sterner, 2000). Moreover, we cannot exclude the possibility that consumers in cryoconite holes may be limited by the lack of nutrients, thus the ingested food composition may shift from its optimum compared to related species from other habitats.

Regarding the $\delta^{13}C$ values, tardigrades and rotifers in our study revealed lower $\delta^{13}C$ than decarbonized cryoconite. This difference is similar to the results described by Almela et al. (2019) and Velázquez et al. (2017) who focused on tardigrades and rotifers from Antarctic microbial mats, but it contrasts with the fundamental literature (Peterson and Fry, 1987; Wada, 2009) as well as with the study of Shaw et al. (2018) who focused on soil in the non-glaciated part of the Taylor Valley (Antarctica). We assume that these variations may be

explained by differences in carbon fractionation on glacier surfaces, differences within tardigrades and rotifers in comparison with freshwater zooplankton and soil microfauna or by the variations in isotopic values of various species which differed in their frequencies among glaciers. The correlation between $\delta^{13}C$ of cryoconite and rotifers may also indicate that rotifer food represents much of the cryoconite organic carbon. Such correlation in tardigrades was not significant, which could be because they potentially consume algae and cyanobacteria that

incorporate $CO_2$ from the atmosphere during photosynthesis with similar $\delta^{13}C$ everywhere. The results presenting the distribution of $\delta^{13}C$ in comparison with differences in $\delta^{15}N$ between tardigrades and rotifers may indicate that rotifers consume DOC originating from extracellular exudates of algae or cyanobacteria (Velázquez et al., 2017), but the source of nitrogen (e.g. bacteria, cyanobacteria and organic detritus) is likely different.

       In comparison with studies focusing on the isotopic composition of consumers from soil and microbial mats in

Antarctica (Almela et al., 2019; Shaw et al., 2018; Velázquez et al., 2017), isotopic composition of tardigrades and rotifers from Arctic cryoconite holes reveals differences in $\delta^{15}N$ as well as in $\delta^{13}C$. Nevertheless, even though studies from Antarctica present different isotopic values, they include important data about relations of tardigrades and rotifers to the main food sources within studied systems, revealing information about the diet of these polar invertebrates and supporting our assumption that both groups probably do not strictly divide their

food sources and therefore a further examination of their gut content is necessary. For example, in Almela et al. (2019), tardigrades were related mostly to a larger fraction of particulate organic matter (POM < 30 μm) composed generally of green algae, instead of rotifers which were related to a smaller fraction of POM (0.5–5 μm) composed generally of bacteria and detritus. In the study of Velázquez et al. (2017), tardigrades were related to cyanobacteria and POM (< 30 μm) and rotifers mostly to cyanobacteria and diatoms. Regarding the

isotopic composition, the closest values to our results were observed in tardigrades and rotifers from soil in Taylor Valley (Shaw et al., 2018) in which these consumers were considered mat grazers.

       It is known that the absolute isotopic composition varies among systems based on various causes, such as differences in the isotopic composition of the nutrient pool (Montoya et al., 1990), seasonal changes in the

community structure (Cifuentes et al., 1988), seasonal variability in isotopic values of the food (Zah et al., 2001)
or due to the effect of temperature on the isotopic fractionation (Bosley et al., 2002; Degens et al., 1968; Hinga et al., 1994; Olive et al., 2003). Thus, our results from cryoconite holes, in which the input of nutrients as well as changes in the community structure of microbes vary during the season (Säwström et al., 2002; Stibal et al., 2008), require further investigation focused on isotopic composition of the gut content in tardigrades and rotifers, their isotopic fractionation and elemental ratio to fully reveal the causes of their different isotopic values.

## 4.2    Variations in isotopic values among glaciers

As shown in the results, the isotopic values among glaciers revealed differences in $\delta^{13}$C of rotifers primarily between Nordenskiöldbreen and Svenbreen. The frequency of consumers on these two glaciers showed a higher abundance of rotifers at Svenbreen and an equal abundance of tardigrades and rotifers at Nordenskiöldbreen. Nordenskiöldbreen also revealed higher amount of presumably microbivorous *P. glacialis* compared to Svenbreen where presumably herbivorous hypsibids dominated.

The differences in $\delta^{13}$C values may indicate specific nutrient requirements of primary producers affected by the variability in spatial characteristics of the glacier surroundings and consequent variations in the nutrient input onto glacier surface (Bagshaw et al., 2013; Hagen et al., 1993). As presented by Post (2002), who focused on freshwater food webs, larger studied lakes evinced higher $\delta^{13}$C values than small lakes suggesting higher occurrence of autochthonous carbon input increasing $\delta^{13}$C of the food web. Based on these findings, we assume that due to its smaller size, Svenbreen may have a higher allochthonous input of nutrients in the form of organic matter from adjacent habitats, which could lower the $\delta^{13}$C because of a longer chain of fractionations discriminating heavier $^{13}$C as it is typical for allochthonous source of carbon (Peterson and Fry, 1987; Post, 2002). Consequently, the depletion in $^{13}$C of consumers on Svenbreen could signify preferential consumption of DOM from the primary production or detritus (Abelson and Hoering, 1961; Iakovenko et al., 2015; Macko and Estep, 1984). Oppositely, consumers from Nordenskiöldbreen and Ebbabreen revealed higher $\delta^{13}$C which could be a result of a larger size of these glaciers and a potential larger component of autochthonous production (Stibal et al., 2010) which uses enriched carbon from atmospheric $CO_2$ (Post, 2002) and has a shorter chain of transformations and discriminations against $^{13}$C during the assimilation of inorganic matter (Michener and Lajtha, 2008). Nevertheless, the observed variations in $\delta^{13}$C among glaciers could also reflect a different proportional representation of herbivorous and other consumers (DeNiro and Epstein, 1978; Michener and Lajtha, 2008), or a dynamical character of sudden processes occurring on the glacial surface including changes in the input of organic and inorganic matter (Chandler et al., 2015; Telling et al., 2012; Wagenbach et al., 1996; Zah et al., 2001). Therefore, further investigations focused on carbon isotopic ratios and fractionation in cryoconite holes are essential.

Regarding the differences in $\delta^{15}$N among glaciers, some samples evinced high presence of cyanobacteria *Leptolyngbya* sp. which may relate to $\delta^{15}$N variations between glaciers due to a higher content of $^{15}$N in the populations of cyanobacteria (Darby and Neher, 2012). However, as described in methods, we were not able to quantify primary producers, thus, our observation may be influenced by inaccuracies caused by the preservation of samples by freezing.

During the analyses of mineral composition of cryoconite, we detected a high amount of amphibole and dolomite on Ebbabreen and Nordenskiöldbreen which are both located in a geologically younger zone of the Billefjorden

Fault Zone compared to Svenbreen located in an older part of the Billefjorden Fault Zone. Considering a higher potential solubility of minerals due to acidic pH of cryoconite holes (4.48–5.9) and differences in mineral composition of cryoconite aggregates among glaciers, the differences in the community structure of microbial communities and consequent isotopic values may be related to the variability in composition of available minerals released by biogeochemical weathering (Barker and Banfield, 1998; Carson et al., 2007; Roberts et al., 2004; Zawierucha et al., 2019c). Moreover, upper parts of Svenbreen were covered by snow during sampling, whereas before and during sampling of Ebbabreen, the air temperature increased to 8.8 °C (according to the meteorological station at Bertilbreen). Therefore, the higher content of $\delta^{15}N$ in these samples could also be caused by the presence of $NO_3^-$ in the meltwater (Hodson et al., 2005).

## 5    Conclusions

This study presents the first description of carbon and nitrogen isotopic values of cryoconite consumers (tardigrades and rotifers) and their potential food. Despite the variability in distribution of isotopic values, we showed that $\delta^{15}N$ differs between tardigrades and rotifers in all samples which points to their different roles in cryoconite trophic network. The $\delta^{13}C$ values revealed variability in their distribution among the taxa as well as between glaciers suggesting that the input and source of carbon among glaciers may differ and influence the isotopic composition of $\delta^{13}C$ in cryoconite as well as in consumers. We also revealed a significant correlation between organic carbon from decarbonized cryoconite and rotifers, which may indirectly indicate that rotifers are more related to cryoconite carbon from bacteria than are tardigrades, which are likely considered to be more herbivorous. Nevertheless, further research is required to elucidate and explain the cryoconite trophic network, the entire diet of the consumers and their contribution to supraglacial nutrient pathways.

## 6 Appendices

**Table A1.** Mineral composition in particular samples analysed by X-Ray diffraction. X letter means presence of the mineral, XX means high presence of the mineral. The sign (–) means that the mineral was not detected.

| Sample | Quartz | Plagioclase | K-Feldspar | Amphibole | Dolomite | Muscovite/Illite | Chlorite |
|--------|--------|-------------|------------|-----------|----------|------------------|----------|
| SL1 | XX | X | X | – | – | XX | X |
| SU1 | XX | X | X | – | – | X | X |
| SL2 | XX | X | X | – | – | XX | XX |
| SU2 | XX | X | X | X | X | XX | XX |
| NL1 | XX | X | X | X | X | XX | XX |
| NU1 | XX | X | X | X | X | XX | XX |
| NL2 | XX | X | X | X | X | XX | XX |
| EL2 | X | X | X | X | XX | XX | XX |
| EU2 | XX | X | X | X | X | XX | XX |

*Code availability*. All codes related to figures and analyses were made in R (version 3. 5. 1.) and are available upon request of the corresponding author.

*Data availability*. All data about isotopic composition, trophic groups composition and mineral composition are available upon request to the corresponding author. Meteorological data from Bertilbreen were kindly provided by Associate Professor Kamil Láska and all requests must be sent to him.

*Author contributions.* JDŽ, TNJ, JT and KZ developed the study design. The field sampling was conducted by TNJ and JDŽ. The stable isotopes analyses were conducted by TNJ, JT and LV. The identification of trophic groups of tardigrades were conducted by TNJ and KZ. The identification of rotifers was conducted by MD. TNJ compiled and processed all presented data and prepared the manuscript contributing revisions from all co-authors.

*Competing interests*. The authors declare that they have no conflict of interest.

*Acknowledgements*. We thank the Research Centre for Radiogenic and Stable Isotopes at the Faculty of Science, Charles University and the Laboratory of X-ray Diffraction of the Institute of Geochemistry, Mineralogy and Mineral Resources, Faculty of Science, Charles University (operator Petr Drahota, Ph.D.) for all support with cryoconite analyses. Furthermore, we would like to thank to Professor J. Middelburg, Professor P. Bartels and all Anonymous Referees for their careful revisions and thorough comments and to Associate Professor Pavel Škaloud, Jakub Štenc, Pavel Pipek, Petra Seifertová, Helena Hubáčková, Jan Soumar, Dan Vondrák, David Novotný and Marina Bukovcak for their specialized help with the determination of primary producers, statistics, lyophilization and manuscript preparation. We also want to thank to Associate Professor Kamil Láska who kindly provided meteorological data from Bertilbreen. Special thanks go to the Cryosphere Ecology Group (cryoeco.eu) and the Department of Ecology at Charles University.

*Financial support*. This research was supported by the Charles University Research Grant GA UK no. 596120 awarded to TNJ; Centre for Polar Ecology and Czech Arctic Polar Infrastructure of University of South Bohemia – Josef Svoboda station at Svalbard (project CzechPolar LM2015078 supported by Ministry of Education, Youth and Sports); Foundation Nadání Josefa, Marie a Zdenky Hlávkových; Mobility Funds of Charles University;

Internationalization Funding of Charles University; Centre for Geosphere Dynamics (UNCE/SCI/006). Studies on organic matter, role of invertebrates and productivity of glacial ecosystems was supported via grant NCN 2018/31/B/NZ8/00198 awarded to KZ.

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
