# Peer review of "Stable isotopic composition of top consumers in Arctic cryoconite holes: revealing divergent roles in a supraglacial trophic network"

_Biogeosciences, 2020_

## Short Comment (SC1) · 14 Apr 2020

This is a very nice paper. Excellent methodology. Thank you for your hard work.

My chief concern is your discussion of the reason for the higher nitrogen content in rotifers than tardigrades and relating that to the potential contribution of bacteria to the cryoconite nitrogen. That seems a bit of a leap. I don't think you established that fully, so this is quite speculative. Right?

I have added comments throughout the manuscript.

[Figure]

Please also note the supplement to this comment:
https://www.biogeosciences-discuss.net/bg-2020-46/bg-2020-46-SC1-supplement.pdf

—————————————————————

[Figure]

**Supplement:**

[revised manuscript text omitted]

---

## Author Comment (AC1) · 20 Apr 2020

Dear Prof. Bartels,

Thank you very much for your thorough comments and corrections. We really appreciate your contribution to improving our study!

Regarding your chief concern. Due to the absence of data about the exact composition of the diet of cryoconite consumers (it is the goal of our current research), we had to operate with various sources of empirical and experimental information about the feeding behaviour of studied groups. The suggestion of a higher $\delta$15N in rotifers

due to a feeding preference of bacteria comes partly from (1) the assumption of a 14N depletion in food for rotifers (bacteria, decomposers) compared to food for tardigrades (mostly algae) (e.g. Kling et al., 1992; Peterson and Fry, 1987; McCutchan et al., 2003) (2) the known higher content of 15N in cyanobacteria due to the fixation of atmospheric nitrogen (e.g. Gu and Alexander, 1993), (3) and from the commonly observed preference for bacterivory in rotifers. We also found a high representation of mostly herbivorous hypsibids which supports the suggestion that the majority of measured tardigrades had fed on algae (e.g. Bryndová et al., 2020). However, all knowledge about the consumers' food we have is based on laboratory experiments, studies of non-glacier species, or experiments with artificial particles. Moreover, as it is evident from the literature, there are many more factors influencing nitrogen stable isotopic ratios in various ecosystems and thus further detailed analyses are needed in cryoconite holes. Therefore, indeed, due to the lack of direct data on composition of the diet our causal explanation of the different isotopic body composition of rotifers remains speculative. We changed the text in order to make it explicit that the mechanism of heavy nitrogen increase in rotifers is an assumption based on indirect empirical data.

Regarding the carnivory of bdelloid rotifers. The only known rotifer predators in cryoconite holes are representatives of the genus Encentrum (Monogonta) which are very rare there. We did not observe any in our study. The Macrotrachella species are as far as we know always microbivores/microfiltrators.

Regarding your last comment about the correlation between cryoconite and rotifer carbon. Indeed, we do not know for sure if bacteria are the major food of rotifers causing this pattern. However, it was a logical conclusion and an indirect evidence based on the assumption of differences in carbon stable isotopic ratios related to the variability in the composition of organic matter between habitats. Moreover, it makes sense in case of tardigrades' consumption of algae which uptake atmospheric $CO_2$ which is the same everywhere.

Thank you very much once more and we will be pleased to continue the discussion if

you have any further questions or concerns.

Yours sincerely,

Tereza Jaroměřská (on behalf of the other authors)

References

Bryndová M., et al. (2020). Dietary preferences and diet effects on life-history traits of tardigrades. Zoological Journal of the Linnean Society, 188(3), 865–877.

Gu B. and Alexander V. (1993). Estimation of N2 fixation based on differences in the natural abundance of 15N among freshwater N2-fixing and non-N2-fixing algae. Oecologia, 96(1), 43–48.

Kling G. W., et al. (1992). Stable isotopes and planktonic trophic structure in arctic lakes. Ecology, 73(2), 561–566.

McCutchan J. H., et al. (2003). Variation in trophic shift for stable isotope ratios of carbon, nitrogen, and sulfur. Oikos, 102(2), 378–390.

Peterson B. J. and Fry B. (1987). Stable isotopes in ecosystem studies. Annual Review of Ecology and Systematics, 485(18), 293–320.

---

## Short Comment (SC2) · 3 May 2020

This paper investigating very important problem about role of animals in functioning of cryoconite hole's ecosystems. Thank you for your work.

I would like to get your attention to more precisely distuinguish occurences of primary producers on different glaciers.

Currently, it is not clear whether e.g. Ancylonema sp. exist on the all glaciers or only in the Ebbabreen or Svenbreen.

This informations can be very useful in the situation of preparing meta-analysis in the

future.

---

## Short Comment (SC3) · 4 May 2020

The article is important to understand the role of the ecological function of invertebrates in cryoconite holes. Please, consider the following suggestions: 1. You should add a short description or reference how you assessed trophic groups of tardigrades. As a pioneer study, this information may be helpful for future research. 2. Based on visual observation of boxplots in groups, variance looks unequal between groups, it may or may not indicate heteroscedasticity. Variance in Svenn seems to be completely different in comparison to the rest of glaciers. In very low sample size it can highly bias result so it can cause Type I error using ANOVA. 3. You should consider changing this

value to a precise number p and r. Value 0.05 as a threshold is arbitrary, thus it can be important in interpreting results, it can be 0.90 or 0.06.
* * *

---

## Author Comment (AC2) · 4 May 2020

Dear Mr. Rozwalak,

thank you very much for reading our work as well as for your comment. We appreciate it very much.

The data presenting the community structure of primary producers on each glacier were not incorporated into the manuscript because the preservation of the samples by freezing has usually a taxon-specific effect on the survival of the cells of phototrophs. Thus, it makes it impossible to provide quantification. For example, the cells of glacial
algae withstand the repeated freeze-thaw cycles only with a low success, whereas cyanobacteria usually have no significant loss in observed cell numbers.

Even though we do have data about the presence or absence of particular species of primary producers on each glacier, we were not able to make a reliable quantitative image of the community structure of primary producers which could help us to elucidate the differences in the stable isotopic composition of cryoconite and invertebrates among glaciers and the differences in trophic groups of tardigrades among glaciers.

Regarding the presence/absence data, all determined species were present on all glaciers except for Leptolyngbya sp., which occurred only on Svenbreen and Ebbabreen.

Thank you very much once more and we will be pleased to continue the discussion if you have any further questions or concerns.

Yours sincerely,

Tereza Jaroměřská (on behalf of the other authors)

---

## Author Comment (AC3) · 15 May 2020

Dear Mr. Buda,

thank you very much for reading our work as well as for your constructive comments. We appreciate it very much.

Regarding your first comment. The description of used methical approach during the division of tardigrades into trophic groups together with the references is described in methods (part 2.5). We considered using of the term dominant feeding groups and trophic groups as a comparable in case of our study. However, due to the improvement

of the comprehensibility of the text, we will unify these terms.

Regarding your second comment. We are aware that due to the limited number of analysed samples is the visualization of results by boxplots favouring a bias result. However, we considered boxplots with the visible values as a convenient and clear presentation of our results. Since our work presents a pioneer study, we took into consideration variation in results in the discussion and highlighted the need of further investigation. Nevertheless, we will consider the mentioning of the risk of used visualization in the legend of the figure.

Regarding the third comment. The r in the correlation between $\delta$13C of decarbonized cryoconite and $\delta$13C of tardigrades is 0.67. We will add this value into results.

Thank you very much once more and we will be pleased to continue the discussion if you have any further questions or concerns.

Yours sincerely,

Tereza Jaroměřská (on behalf of the other authors)

---

## Referee Comment (RC2) · Anonymous Referee #2 · 8 Jun 2020

Dear authors, dear editor,

My apologies for the long delay in reviewing your manuscript!

I have now read your manuscript and provide my comments. First, it is important to state that I am less familiar with gletsjer ecology and geology, and that it is the first paper I read about cryoconite holes.

Having said that, while I feel that the paper presents interesting and novel data, I also strongly feel that it lacks conclusiveness to warrant publication in a top journal like Biogeosciences. My main reason for this somewhat harsh conclusion is that the results presented on stable isotope signatures of tardigrades, rotifers and cryoconite 'sedi-

ments+OM' of three gletsjer systems can only be described and compared, but that virtually all data needed to explain the observed patterns are lacking. I perfectly understand that the authors set out to do a pioneering study on the trophic ecology of the most abundant consumers in these holes, but their explanations for the obtained results remain very speculative for lack of measurements of both resources and subsidies. I would not object to the data being published in a more specialised journal, but in Biogeosciences, I expect to read papers which provide mechanistic understanding of patterns and processes, and while there is relevant discussion about this, the obvious absence of the necessary data for this prohibits any substantial conclusions. Isotopic signatures of nutrient and OM subsidies are missing, and so are isotopic signatures of primary producers vs detritus vs heterotrophic bacteria. I perfectly understand that these are not easy to obtain, but it was foreseeable that such data would be required to explain observed patterns, and methods exist for determining each of the above factors.

Apart from this general appreciation, I also have some more specific contents. In the introduction, I feel the authors should do an extra effort to make this manuscript better accessible and therefore more interesting for readers who are not familiar with the cryoconite holes. Aspects I wondered about, not knowing these specific systems, is what size and depth range such cryoconite holes have and how common they are, or better still, what % of a gletsjer's surface they make up. It is one element I would need to answer the "so what" question: what could be the quantitative importance of these 'miniature' ecosystems in the ecology and biogeochemistry of gletsjers?

I would also have liked to read more clearly structured information about the different organic matter resources – both autochthonous and allochthonous – that tend to be present in cryoconite holes. Particularly on the primary producers, info is rather minimal.

And as you were comparing three different gletsjer systems, I would have hoped that you had some clear rationale for choosing precisely these three systems, which would

then translate in one or more clear hypotheses about what (different) patterns to expect. Now, there are no very clear goals , questions or hypotheses about such expected patterns, and I am left with the question: since you sampled glaciers in three different settings, how did you expect their food webs and fuelling resources to differ?

In lines 44-47, you are mentioning a correlation between the size distribution of primary producers and the community composition of consumers. In what way, and what is the relevance here?

The materials and methods section is generally well described, but I was rather surprised to read the following final sentence of the conclusions section: "Another outcome of this study is the introduction of a modified technique of sample preparation avoiding procedures such as sugar centrifugation or oven drying." While reading M&M, I did notice some minor differences in sample elutriation and sorting procedures compared to common practices, but nowhere did I see a clear statement about an important novel methodological approach to treating samples.

Given the subtitle 'community structure', I had somehow hoped to read a species or genus-level analysis of consumer communities, yet in the end, rotifers were just treated as rotifers, and tardigrades were largely left without a name and assigned to feeding groups, which are well known to be of very limited relevance to describe the actual feeding behaviours of tardigrades, particularly with respect to their potential to feed on microalgae. Similarly, whilst the authors explain that they identified cyanobacteria and algae, we later learn that they looked at roughly 10% of these primary producers, because sample preservation could have strongly biased actual community composition. So there is no info on community composition, only some more anecdotal statements as to certain abundantly present Cyanobacteria and algae. This is probably also why there is no statistical analysis of differences in community composition of either consumers or primary producers. Incidentally, I wondered why both a Spearman rank and a Pearson product-moment correlation were tested. I would expect that if the data fit the assumptions for parametric tests, one would choose the latter, and if not, the

former.

The results are generally well presented. I have only some specific suggestions. 1) I would have preferred to see absolute abundances of tardigrades and rotifers in table 1 instead of a general dominance-indication. That dominance indication is not very scientific, and it does not provide any relevant info on potential differences in abundance of consumer groups among gletsjer systems. Actually, since you sampled sufficient replicates in 2 out of 3 gletsjers, you could statistically underpin whether tardigrades were more or less abundant than rotifers in a specific gletsjer system. 2) Figure only shows the correlation between cryoconite delta13C and rotifer delta13C. That of tardigrades is not shown because it was not statistically significant. I would then at least like to see the r and actual p-value for the tardigrade correlation, as well as an estimate of the slope of the regression for both rotifers and tardigrades. That would allow me a clearer picture on whether tardigrades had a somewhat less clear but still generally very similar pattern as rotifers, or whether their patterns differed importantly. 3) I would have started the results section with what is now 3.2. Cryoconite composition. 4) Section 3.3: as mentioned above, table 1 should provide absolute abundances of consumers, and it should also provide info on the proportion of dormant consumers! 5) I would be interested to read in one or two sentences to what extent other meiofauna or small invertebrates than tardigrades and rotifers were present (any Nematoda, Copepoda, Ostracoda, . . .?).

Discussion: At the end of the first discussion §, you finally provide some 'expectation', but it is not clear what this expectation is based upon.

In the relatively lengthy and speculative discussion, I read virtually nothing about temperature effects on isotopic fractionation between consumers and resources. I believe that there is some relevant literature on that, and it should have been included here.

In section 4.2., dissolved organic matter suddenly becomes an important candidate food for these consumers. That may well be, but what results is this based upon here?

Finally, there are a few instances where isotope 'terminology' is used in a somewhat sloppy way: 1) Line 71: "preferential excretion of $\delta$14N" should be "preferential excretion of (light) 14N". 2) Line 250: "rotifers revealed higher values of 15N isotope" should be "rotifers had higher $\delta$15N" or "rotifers had heavier stable nitrogen isotopic signatures'. 3) Line 251: "potential differences in $\delta$15N composition" should be "potential differences in N isotope ratios/signatures".

---

## Author Comment (AC5) · 26 Jun 2020

Dear Referee and dear Prof. Middelburg,

thank you very much for your very useful and thorough comments and corrections. We appreciate your contribution to the improvement of our study very much.

We will modify the text of the manuscript accordingly and we also provided detailed answers to your comments below (comments are in bold and our responses are in regular).

1. **Having said that, while I feel that the paper presents interesting and novel data, I also strongly feel that it lacks conclusiveness to warrant publication in a top journal like Biogeosciences. My main reason for this somewhat harsh conclusion is that the results presented on stable isotope signatures of tardigrades, rotifers and cryoconite of three gletsjer systems can only be described and compared, but that virtually all data needed to explain the observed patterns are lacking.**

We are aware that this manuscript is a pioneer descriptive study presenting the first view on stable isotopic composition of consumers and organic matter on the glacier surface with all limitations which it may pose on the generalization of our conclusions. Conclusions include parts where our results were overinterpreted and where our suggestions go beyond the safe ground provided by our data. We will change these parts accordingly.

2. **Isotopic signatures of nutrient and OM subsidies are missing, and so are isotopic signatures of primary producers vs detritus vs heterotrophic bacteria. I perfectly understand that these are not easy to obtain, but it was foreseeable that such data would be required to explain observed patterns, and methods exist for determining each of the above factors.**

We are aware that isotopic signatures of primary producers (ideally cyanobacteria and algae separately), heterotrophic bacteria and detritus in cryoconite holes are necessary to understand the trophic structure and processes leading to presented differences in isotopic signatures of cryoconite tardigrades and rotifers. However, the main purpose of our study was not to provide detailed description of trophic relationships and pathways in cryoconite holes but rather to obtain a general view on the stable isotopic composition of supraglacial tardigrades and rotifers and find out whether it is worth keeping this direction in further studies or not. The chosen approach exposed that the food sources of both consumers probably differ even though feeding experiments showed their ability to consume the same food (Střítecká and Devetter, 2015). We agree that there are other methods such as SIMS which may solve these problems. However, we did not choose such methods due to the quantity of material necessary for our purpose. We feel that we were not explicit in this subject and will modify the discussion accordingly.

3. **I feel the authors should do an extra effort to make this manuscript better accessible and therefore more interesting for readers who are not familiar with the cryoconite holes. Aspects I wondered about, not knowing these specific systems, is what size and depth range such cryoconite holes have and how common they are, or better still, what % of a gletsjer's surface they make up.**

We appreciate this suggestion very much. We will modify the abstract and the introduction to make the study better accessible for readers less familiar with the topic.

The diameter and depth of cryoconite holes are noticeably variable, but both usually range from a few centimetres to tens of centimetres (Gerdel and Drouet, 1960; Fountain et al., 2004; Zawierucha et al., 2019).

Cryoconite holes usually cover around 7 % of the ablation zone of glaciers (the area with an ice loss exceeding its increase), which still remains a relevant portion of the planetary surface (Bøggild et al., 2010; Fountain et al., 2004; Stibal et al., 2012a; Zemp et al., 2008). Their distribution and structure (e.g. size and depth) depend on many factors such as the size of the ablation zone, the insolation on the glacier surface, optical properties of the ice, temperature, sediment input (mainly airborne) and output (mainly with meltwater), type of sediment, type of adjacent area (terrestrial, marine) and the topography of the glacier surface.

The fact that the supraglacial sediment has an active biotic component contributing to its distribution and physical and chemical properties, makes it an important factor that has to be considered when assessing the functioning of biogeochemical processes in glaciated catchments and the impact on e.g. understanding ecological succession in deglaciated areas and the change of marine ecosystems in the Arctic.

**4. It is one element I would need to answer the "so what" question: what could be the quantitative importance of these 'miniature' ecosystems in the ecology and biogeochemistry of gletsjers?**

We appreciate this comment and we will modify the introduction accordingly.

As reported by Cameron et al. (2012) and Hodson et al. (2008) and explained previously here, cryoconite holes are the most nutrient-rich and biologically active habitats within the supraglacial environment. Therefore, their impact on glacier ecosystems nutrient pathways (e.g. carbon, nitrogen, and other microelements) and on downstream ecosystems is a key component to understanding the glacial ecosystems functioning (Anesio et al., 2010; Bagshaw et al., 2013; Telling et al., 2011). For example, the rate of photosynthesis in cryoconite holes is comparable with those in arctic polar lakes and consequently the rate of respiration and utilization of organic matter is very high (Hodson et al., 2008). Thus, cryoconite holes form an important net carbon sink or source in polar ecosystems which depend on the balance between autotrophic and heterotrophic production (Stibal et al., 2012b). Moreover, due to its high biological activity, cryoconite holes efficiently retain nutrients on the glacier surface (Bagshaw et al., 2013) and accumulated matter can consequently provide a source of important nutrients into adjacent areas where it is transported with the meltwater (Anesio et al., 2010; Porazinska et al., 2004).

Another importance of cryoconite holes comes from the dark colour of cryoconite granules which alters the albedo of the glacier surface and can lead to an increase in the surface melt and eventually to the expansion of the ablation zone (Ryan et al., 2018; Takeuchi et al., 2001). Due its accumulative nature, cryoconite holes are also used for the exploration and tracking of contamination (e.g. black carbon, radionuclides) in polar areas (e.g. Łokas et al., 2016).

Tardigrades and rotifers inhabiting cryoconite holes are most likely an important factor shaping the community of primary producers by grazing and nutrient recycling (Elser and Urabe, 1999; Vonnahme et al., 2016). Especially when considering their feeding rate (Střítecká and Devetter, 2015). The closest similar studies about stable isotopic composition of tardigrades and rotifers focused only on microbial mats and soil from Antarctica (Almela et al., 2019; Shaw et al., 2018; Velázquez et al., 2017). Due to the dominance of prokaryotic biomass which was in the main scope of

most of previous studies from supraglacial habitats, the ecology of cryoconite tardigrades and rotifers and their influence on the supraglacial system functioning is currently unexplored.

**5. I would also have liked to read more clearly structured information about the different organic matter resources – both autochthonous and allochthonous – that tend to be present in cryoconite holes.**

Thank you for this suggestion. We will modify the introduction accordingly.

Regarding allochthonous sources of organic matter, it is mostly dust from adjacent areas or snow which is transported to cryoconite holes with meltwater or wind. The components of allochthonous organic matter are mostly death organic matter, snow algae, remains of plants and organisms (Takeuchi et al., 2001) and bird guano (Vonnahme et al., 2016; Žárský et al., 2013).

The autochthonous sources of organic matter are mostly of microbial origin (from cryoconite bacteria, cyanobacteria, and algae) (e.g. Telling et al., 2012).

**6. Now, there are no very clear goals, questions or hypotheses about such expected patterns, and I am left with the question: since you sampled glaciers in three different settings, how did you expect their food webs and fuelling resources to differ?**

The heterogeneity of three chosen glaciers (size, adjacent areas, geological setting, distance from the sea) should demonstrate possible regional variability in stable isotopic composition of its cryoconite consumers and organic matter among various glaciers in central Svalbard. We expected that different geomorphological characteristics will be reflected in the input of organic matter (e.g. Svenbreen is surrounded by steep slopes) and thus in the composition of their consumers. Nevertheless, our selection should not be understood as a sufficient source of information for hypothesis-driven analysis of controls of the stable isotopic composition in our study.

Based on previous studies we knew that the composition of biota tends to differ between glaciers (e.g. Cameron et al., 2012; Edwards et al., 2013a; Edwards et al., 2013b). The main objective of our study was to use stable isotopic analyses as a tool which could reveal if the food source of cryoconite consumers differ and therefore give an indication whether tardigrades and rotifers can be reasonably expected to influence the cryoconite stoichiometry.

Thank you for this comment, we agree that we did not mark the goals of our study clearly and we will modify the text accordingly.

**7. In lines 44-47, you are mentioning a correlation between the size distribution of primary producers and the community composition of consumers. In what way, and what is the relevance here?**

Vonnahme et al. (2016) described that the abundance of rotifers correlates with larger and smaller microalgae (Chlorococcales and Zygnematales) concentrations and the abundance of tardigrades only with larger Zygnematales concentration. This study also revealed that the length of trichomes of Oscillatoriales (cyanobacteria) negatively correlates with the abundance of filtrating rotifers. We felt that it is important to mention these findings because it indicates that grazing has likely an impact on

the structure of primary producers in cryoconite holes and by that presumably contributes to cryoconite nutrient levels.

We appreciate this comment and we will modify the text to make the meaning more explicit.

8. **I was rather surprised to read the following final sentence of the conclusions section: "Another outcome of this study is the introduction of a modified technique of sample preparation avoiding procedures such as sugar centrifugation or oven drying." While reading M&M, I did notice some minor differences in sample elutriation and sorting procedures compared to common practices, but nowhere did I see a clear statement about an important novel methodological approach to treating samples.**

Thank you for this comment, we agree that we did not describe precisely why we modified the technique. We are also aware that our methodical approach did not present a fully novel method. We will modify the text accordingly.

The commonly used methods were modified because cryoconite invertebrates live in a very specific environment and we wanted to avoid alteration of their chemical composition during the preparation for isotopic analyses. Therefore, we chose the lyophilization instead of oven drying and we wanted to avoid any added component which could potentially contaminate our samples.

9. **Given the subtitle 'community structure', I had somehow hoped to read a species or genus-level analysis of consumer communities, yet in the end, rotifers were just treated as rotifers, and tardigrades were largely left without a name and assigned to feeding groups, which are well known to be of very limited relevance to describe the actual feeding behaviours of tardigrades, particularly with respect to their potential to feed on microalgae.**

Thank you for this comment, we will replace the community structure by the consumers composition and will change the text accordingly.

We are also aware that community structure description lacks detailed information about species composition. Regarding rotifers, the identification depends on the visibility of their coronal cilia which only actively filtrating individuals display. We did not observe a sufficient number of filtrating rotifers during the collection of rotifers for analyses and therefore other identification except few individuals identified as *Macrotrachella* sp. and *Adineta* sp. was impossible.

Regarding the species composition of tardigrades, the representative number of individuals was collected, mounted, and determined apart from individuals intended for analyses. We found *Pilatobius* sp., *Hypsibius* sp., *Hypsibius* cf. *dujardini*, *Isohypsibius* sp. and very rarely *Cryoconicus kaczmareki. Pilatobius* sp. was in the past identified as *P. recamieri* which inhabit Arctic tundra commonly. However, utilization of DNA barcoding revealed that suggested *P. recamieri* is a new species for science and it is not formally named yet. *Hypsibius* contains, according to DNA, few cryptic lines. *Grevenius* (previously identified as *Isohypsibius* sp.) is also new for science and not formally named. Moreover, because some species are cryptic and identified only based on DNA (morphologically similar), they most probably use the same food source. Due to the lack of information about the diet of particular species living in cryoconite holes, we chose the trophic group division based on already published knowledge about feeding behaviours and feeding apparatus morphology of related species and on the personal communication with specialists culturing relative species.

**10. So, there is no info on community composition, only some more anecdotal statements as to certain abundantly present Cyanobacteria and algae. This is probably also why there is no statistical analysis of differences in community composition of either consumers or primary producers.**

The data presenting the community structure of primary producers on each glacier were not incorporated into the manuscript because the preservation of the samples by freezing has usually a taxon-specific effect on the survival of the cells of phototrophs. Thus, it makes it impossible to provide reliable quantification comparable between taxa. For example, the cells of glacial algae withstand the repeated freeze-thaw cycles only with a low success, whereas cyanobacteria usually have no significant loss in observed cell numbers. Even though we agree that the information lacks further data, we felt that it is important to mention at least all information we were able to get from our samples. Due to these reasons, we excluded the data on primary producers from statistical analyses.

The community composition of consumers could not be statistically analysed since were not able to provide sufficient data about the species composition of rotifers among glaciers.

**11. Incidentally, I wondered why both a Spearman rank and a Pearson product-moment correlation were tested. I would expect that if the data fit the assumptions for parametric tests, one would choose the latter, and if not, the former.**

We had to use both tests because some data did not have a normal distribution. However, we were not explicit in the methods that the tests have not been used both on the same data. We will modify the text to make the meaning clear.

**12. I would have preferred to see absolute abundances of tardigrades and rotifers in table 1 instead of a general dominance-indication. That dominance indication is not very scientific, and it does not provide any relevant info on potential differences in abundance of consumer groups among gletsjer systems. Actually, since you sampled sufficient replicates in 2 out of 3 gletsjers, you could statistically underpin whether tardigrades were more or less abundant than rotifers in a specific gletsjer system.**

Thank you for this suggestion. We will modify the table and add the statistics into the text.

We chose visualisation with the dominance-indication due to an easier description and orientation in results for readers since the total number of collected animals from which was the dominance estimated is much higher than the number of animals necessary for analyses. Many counted animals were analysed in pilot experiments with no results. We have changed the table accordingly (presented below) and we will make the description more explicit.

| Glacier | Tardigrades | | Rotifers | | Cryoconite | | Dominances |
|---|---|---|---|---|---|---|---|
| | $\delta^{15}N$ | $\delta^{13}C$ | $\delta^{15}N$ | $\delta^{13}C$ | $\delta^{15}N$ | $\delta^{13}C*$ | T:R |
| Sven | −3.55 | −23.75 | −1.57 | −27.09 | −2.20 | −22.65 | 1030:627 |
| | −4.04 | −26.36 | −1.81 | −26.69 | −2.17 | −22.08 | 1062:2184 |
| | −3.52 | −26.33 | −2.04 | −31.16 | −2.24 | −24.09 | 1181:1381 |

| | | | | | | | |
|---|---|---|---|---|---|---|---|
| | −4.39 | −26.45 | −1.04 | −28.53 | −1.64 | −23.25 | 150:1021 |
| Nordenskiöld | −4.39 | −22.91 | −2.20 | −23.36 | −3.58 | −21.56 | 1172:1091 |
| | −3.45 | −23.30 | −1.72 | −23.25 | −2.30 | −20.56 | 1254:1355 |
| | −3.76 | −22.58 | −2.30 | −23.02 | −2.98 | −22.52 | |
| | −4.15 | −24.78 | −3.02 | −25.19 | | | 1225:1341 |
| | −3.93 | −25.11 | | | | | |
| Ebba | −1.73 | −24.19 | 0.69 | −26.32 | −2.30 | −22.24 | 1086:2184 |
| | −4.85 | −25.22 | −2.38 | −25.13 | −4.29 | −23.31 | 2280:594 |
| | −4.33 | −25.15 | | | | | |

**13. Figure only shows the correlation between cryoconite delta13C and rotifer delta13C. That of tardigrades is not shown because it was not statistically significant. I would then at least like to see the r and actual p-value for the tardigrade correlation, as well as an estimate of the slope of the regression for both rotifers and tardigrades.**

Thank you very much for this suggestion, we will add all information into the text. The r in the correlation between $\delta^{13}C$ of decarbonized cryoconite and $\delta^{13}C$ of tardigrades is 0.67 and the *p*-value is 0.07. We will also add this value and suggested table (see below) into the results.

[Figure]

**14. I would have started the results section with what is now 3.2. Cryoconite composition.**

We appreciate this suggestion and we will incorporate the section 3.2 into the section 3.1.

**15. Section 3.3: as mentioned above, table 1 should provide absolute abundances of consumers, and it should also provide info on the proportion of dormant consumers!**

The proportion of dormant consumers is not possible to count during the sample preparation procedure because we use melted material and dormant tardigrades and rotifers are waking up during the whole time of the collecting for isotopic analyses.

**16. I would be interested to read in one or two sentences to what extent other meiofauna or small invertebrates than tardigrades and rotifers were present (any Nematoda, Copepoda, Ostracoda, . . .?).**

Based on current knowledge, Arctic cryoconite holes are inhabited exclusively by tardigrades and rotifers. Currently, we have data from 20 glaciers from Svalbard which indicate that no other animals inhabit cryoconite holes in this region. For our investigation, we used material from many seasons which was analysed by students and experienced researchers and no other metazoans have been found. Moreover, only few groups of other animals inhabit cryoconite holes worldwide and most of these species are endemic. Crustaceans (Copepoda) were found only in Himalaya, insects (Chironomidae and Plecoptera) have been found in the Himalaya and Patagonia, recently mites have been found on one Antarctic glacier. Our unpublished meta-analysis suggests that tardigrades and rotifers are the most common cryoconite animals. We will correct the introduction and we will underline that exclusively rotifers and tardigrades play roles as apex consumer of cryoconite holes in the Arctic. We will cite adequate references as well.

**17. At the end of the first discussion §, you finally provide some 'expectation', but it is not clear what this expectation is based upon. In the relatively lengthy and speculative discussion, I read virtually nothing about temperature effects on isotopic fractionation between consumers and resources.**

Thank you for this suggestion, we will add the information into the discussion. We will also modify the discussion to make its meaning explicit.

The temperature in cryoconite holes is very stable (around 0 °C) and the fluctuation usually do not exceed tenths of °C (e.g. Säwström et al., 2002; Zawierucha et al., 2019). Any increase in the temperature of the sediment is efficiently buffered by ice at 0 °C because any added heat will lead to melting instead. Publications focusing on the temperature effect on the isotopic fractionation usually describe that the changes in fractionation begin with differences higher than 2 °C (e.g. Bosley et al., 2002; Degens et al., 1968; Hinga et al., 1994; Olive et al., 2003).

**18. In section 4.2., dissolved organic matter suddenly becomes an important candidate food for these consumers. That may well be, but what results is this based upon here?**

This suggestion is based on the description and explanation of observed distribution of $\delta^{13}C$ in consumers and DOC from cyanobacterial exudates in the study of Velázquez et al. (2017). This study focused, apart from microbial trophic interaction, on isotopic composition of tardigrades and rotifers from Antarctica as well. However, this is only one of more possible explanations. Therefore, we will modify the text to make this consideration more explicit.

**19. 1) Line 71: "preferential excretion of δ14N" should be "preferential excretion of (light) 14N". 2) Line 250: "rotifers revealed higher values of 15N isotope" should be "rotifers had higher δ15N" or "rotifers had heavier stable nitrogen isotopic signatures'. 3) Line 251: "potential differences in δ15N composition" should be "potential differences in N isotope ratios/signatures".**

Thank you very much for these corrections. We agree with all suggestions and we will modify the text accordingly.

Thank you very much once more and we will be pleased to continue the discussion if you have any further questions or concerns.

Yours sincerely,

Tereza Jaroměřská

(on behalf of the other authors)

**References**

[revised manuscript text omitted]

---

## Author Response (AR1)

Dear prof. Middelburg, dear referees,

we are submitting a revised version of our manuscript. Thank you for all reviews, we believe that they helped to improve the quality of our manuscript. We tried to take them into account and incorporate all suggestions.

Here, we present all the changes and the answers to referees and scientific community.

Yours sincerely,

Tereza Jaroměřská

**Point-by-point response to the reviews**

**Responses to prof. Bartels**

**1. My chief concern is your discussion of the reason for the higher nitrogen content in rotifers than tardigrades and relating that to the potential contribution of bacteria to the cryoconite nitrogen. That seems a bit of a leap. I don't think you established that fully, so this is quite speculative. Right?**

Due to the absence of data about the exact composition of the diet of cryoconite consumers (it is the aim of our current research), we had to operate with various sources of empirical and experimental information about the feeding behaviour of studied groups. The suggestion of a higher $\delta^{15}N$ in rotifers due to a feeding preference of bacteria comes partly from (1) the assumption of a $^{14}N$ depletion in food for rotifers (bacteria, decomposers) compared to food for tardigrades (mostly algae) (e.g. Kling et al., 1992; Peterson and Fry, 1987; McCutchan et al., 2003) (2) the known higher content of $^{15}N$ in cyanobacteria due to the fixation of atmospheric nitrogen (e.g. Gu and Alexander, 1993), (3) and from the commonly observed preference for bacterivory in rotifers. We also found a high representation of mostly herbivorous hypsibids which supports the suggestion that the majority of measured tardigrades had fed on algae (e.g. Bryndová et al., 2020). However, all knowledge about the consumers' food we have is based on laboratory experiments, studies of non-glacier species, or experiments with artificial particles. Moreover, as it is evident from the literature, there are many more factors influencing nitrogen stable isotopic ratios, in various ecosystems, including the fractionation within the body of consumer based on the isotopic composition of the food (Aberle and Malzahn, 2007), and thus further detailed analyses are needed in cryoconite holes.

Therefore, indeed, due to the lack of direct data on composition of the diet our causal explanation of the different isotopic body composition of rotifers remains speculative. We changed the text in order to make it explicit that the mechanism of heavy nitrogen increase in rotifers is an assumption based on indirect empirical data.

**2. Abstract (line 14): Explain dynamic nature: you mean the food webs themselves are dynamic?**

By "dynamic nature" we meant the dynamic nature of the formation and functioning of cryoconite holes. We changed the sentence to make this meaning more explicit.

**3. Discussion (line 244): Strange to say various causes then only give one example. Need to specify other causes.**

The sentence was modified and we added other examples.

**4. Discussion (line 265–270): Isn't it possible that some of the rotifers are carnivorous (or at least omnivorous)? That would also explain the pattern.**

The only known rotifer predators in cryoconite holes are representatives of the genus *Encentrum* (Monogonta) which are very rare there. We did not observe any in our study. The *Macrotrachella* species are as far as we know always microbivores/microfiltrators.

**5. Discussion (line 277): Unclear what you mean by representative. You mean bacteria is the predominant determinant of $\delta^{13}C$?**

We do not know for sure if bacteria are the major food of rotifers causing this pattern. However, it was a logical conclusion based on the assumption and an indirect evidence of differences in carbon stable isotopic ratios related to the variability in the composition of organic matter between

habitats. Moreover, it makes sense in case of tardigrades' consumption of algae which uptake atmospheric $CO_2$ which is the same everywhere.

However, we modified the paragraph in order to make its meaning clear.

**6. Conclusions (line 334–335): This seems like a big logical jump. You don't really know that bacteria are the major contributor to C, so this becomes circular reasoning.**

We agree and we modified the conclusion and discussion to highlight that this suggestion is an indirect evidence which needs further investigation focused on the particular diet of cryoconite consumers.

**Responses to Referee #2**

**1. Having said that, while I feel that the paper presents interesting and novel data, I also strongly feel that it lacks conclusiveness to warrant publication in a top journal like Biogeosciences. My main reason for this somewhat harsh conclusion is that the results presented on stable isotope signatures of tardigrades, rotifers and cryoconite of three gletsjer systems can only be described and compared, but that virtually all data needed to explain the observed patterns are lacking.**

We are aware that this manuscript is a pioneer descriptive study presenting the first view on stable isotopic composition of consumers and organic matter on the glacier surface with all limitations which it may pose on the generalization of our conclusions. Conclusions include parts where our results were overinterpreted and where our suggestions go beyond the safe ground provided by our data. We changed these parts accordingly.

**2. Isotopic signatures of nutrient and OM subsidies are missing, and so are isotopic signatures of primary producers vs detritus vs heterotrophic bacteria. I perfectly understand that these are not easy to obtain, but it was foreseeable that such data would be required to explain observed patterns, and methods exist for determining each of the above factors.**

We are aware that isotopic signatures of primary producers (ideally cyanobacteria and algae separately), heterotrophic bacteria and detritus in cryoconite holes are necessary to understand the trophic structure and processes leading to presented differences in isotopic signatures of cryoconite tardigrades and rotifers. However, the main purpose of our study was not to provide detailed description of trophic relationships and pathways in cryoconite holes but rather to obtain a general view on the stable isotopic composition of supraglacial tardigrades and rotifers and find out whether it is worth keeping this direction in further studies or not. We agree that there are other methods such as SIMS which may solve these problems. However, we did not choose such methods due to the quantity of material necessary for our purpose. We feel that we were not explicit in this subject and modified the text accordingly.

**3. I feel the authors should do an extra effort to make this manuscript better accessible and therefore more interesting for readers who are not familiar with the cryoconite holes. Aspects I wondered about, not knowing these specific systems, is what size and depth range such cryoconite holes have and how common they are, or better still, what % of a gletsjer's surface they make up.**

The diameter and depth of cryoconite holes are noticeably variable, but both usually range from a few centimetres to tens of centimetres (Gerdel and Drouet, 1960; Fountain et al., 2004; Zawierucha et al., 2019a).

Cryoconite holes usually cover around 7 % of the ablation zone of glaciers (the area with an ice loss exceeding its increase), which still remains a relevant portion of the planetary surface (Bøggild et al., 2010; Fountain et al., 2004; Stibal et al., 2012a; Zemp et al., 2008). Their distribution and structure (e.g. size and depth) depend on many factors such as the size of the ablation zone, the insolation on the glacier surface, optical properties of the ice, temperature, sediment input (mainly airborne) and output (mainly with meltwater), type of sediment, type of adjacent area (terrestrial, marine) and the topography of the glacier surface.

The fact that the supraglacial sediment has an active biotic component contributing to its distribution and physical and chemical properties, makes it an important factor that has to be considered when assessing the functioning of biogeochemical processes in glaciated catchments and the impact on e.g. understanding ecological succession in deglaciated areas and the change of marine ecosystems in the Arctic.

We modified the abstract and the introduction to make the study better accessible for readers less familiar with the topic.

**4. It is one element I would need to answer the "so what" question: what could be the quantitative importance of these 'miniature' ecosystems in the ecology and biogeochemistry of gletsjers?**

As reported by Cameron et al. (2012) and Hodson et al. (2008), cryoconite holes are the most nutrient-rich and biologically active habitats within the supraglacial environment. Therefore, their impact on glacier ecosystems nutrient pathways (e.g. carbon, nitrogen, and other microelements) and on downstream ecosystems is a key component to understanding the glacial ecosystems functioning (Anesio et al., 2010; Bagshaw et al., 2013; Telling et al., 2011). For example, the rate of photosynthesis in cryoconite holes is comparable with those in arctic polar lakes and consequently the rate of respiration and utilization of organic matter is very high (Säwström et al., 2002). Thus, cryoconite holes form an important net carbon sink or source in polar ecosystems which depend on the balance between autotrophic and heterotrophic production (Stibal et al., 2012b). Moreover, due to its high biological activity, cryoconite holes efficiently retain nutrients on the glacier surface (Bagshaw et al., 2013) and accumulated matter can consequently provide a source of important nutrients into adjacent areas where it is transported with the meltwater (Anesio et al., 2010; Porazinska et al., 2004).

Another importance of cryoconite holes comes from the dark colour of cryoconite granules which alters the albedo of the glacier surface and can lead to an increase in the surface melt and eventually to the expansion of the ablation zone (Ryan et al., 2018; Takeuchi et al., 2001). Due to its accumulative nature, cryoconite holes are also used for the exploration and tracking of contamination (e.g. black carbon, radionuclides) in polar areas (e.g. Łokas et al., 2016).

Tardigrades and rotifers inhabiting cryoconite holes are most likely an important factor shaping the community of primary producers by grazing and nutrient recycling (Elser and Urabe, 1999; Vonnahme et al., 2016). Especially when considering their feeding rate (Střítecká and Devetter, 2015). The closest similar studies about stable isotopic composition of tardigrades and rotifers focused only on microbial mats and soil from Antarctica (Almela et al., 2019; Shaw et al., 2018; Velázquez et al., 2017). Due to the dominance of prokaryotic biomass which was in the main scope of most of previous studies from supraglacial habitats, the ecology of cryoconite tardigrades and rotifers and their influence on the supraglacial system functioning is currently unexplored.

We modified the introduction and added information about the importance of cryoconite holes.

**5. I would also have liked to read more clearly structured information about the different organic matter resources – both autochthonous and allochthonous – that tend to be present in cryoconite holes.**

Regarding allochthonous sources of organic matter, it is mostly dust from adjacent areas or snow which is transported to cryoconite holes with meltwater or wind. The components of allochthonous organic matter are mostly death organic matter, snow algae, remains of plants and organisms (Takeuchi et al., 2001) and bird guano (Vonnahme et al., 2016; Žárský et al., 2013).

The autochthonous sources of organic matter are mostly of microbial origin (from cryoconite bacteria, cyanobacteria, and algae) (e.g. Telling et al., 2012).

We added all above-mentioned into the introduction.

**6. Now, there are no very clear goals, questions or hypotheses about such expected patterns, and I am left with the question: since you sampled glaciers in three different settings, how did you expect their food webs and fuelling resources to differ?**

The heterogeneity of three chosen glaciers (size, adjacent areas, geological setting, distance from the sea) should demonstrate possible regional variability in stable isotopic composition of its cryoconite consumers and organic matter among various glaciers in central Svalbard. We expected that different geomorphological characteristics will be reflected in the input of organic matter (e.g. Svenbreen is surrounded by steep slopes) and thus in the composition of their consumers. Nevertheless, our selection should not be understood as a sufficient source of information for hypothesis-driven analysis of controls of the stable isotopic composition in our study.

Based on previous studies we knew that the composition of biota tends to differ between glaciers (e.g. Cameron et al., 2012; Edwards et al., 2013a; Edwards et al., 2013b). The main objective of our study was to use stable isotopic analyses as a tool which could reveal if the food source of cryoconite consumers differ and therefore give an indication whether tardigrades and rotifers can be reasonably expected to influence the cryoconite stoichiometry.

We modified the introduction and added above-mentioned into the text.

**7. In lines 44-47, you are mentioning a correlation between the size distribution of primary producers and the community composition of consumers. In what way, and what is the relevance here?**

Vonnahme et al. (2016) described that the abundance of rotifers correlates with larger and smaller microalgae (Chlorococcales and Zygnematales) concentrations and the abundance of tardigrades only with larger Zygnematales concentration. This study also revealed that the length of trichomes of Oscillatoriales (cyanobacteria) negatively correlates with the abundance of filtrating rotifers. We felt that it is important to mention these findings because it indicates that grazing has likely an impact on the structure of primary producers in cryoconite holes and by that presumably contributes to cryoconite nutrient levels.

We modified the text to make the meaning more explicit.

**8. I was rather surprised to read the following final sentence of the conclusions section: "Another outcome of this study is the introduction of a modified technique of sample preparation avoiding procedures such as sugar centrifugation or oven drying." While reading M&M, I did notice some minor differences in sample elutriation and sorting procedures compared to common practices,**

**but nowhere did I see a clear statement about an important novel methodological approach to treating samples.**

The commonly used methods were modified because cryoconite invertebrates live in a very specific environment and we wanted to avoid alteration of their chemical composition during the preparation for isotopic analyses. Therefore, we chose the lyophilization instead of oven drying and we wanted to avoid any added component which could potentially contaminate our samples.

We agree that we did not describe precisely why we modified the technique. We are also aware that our methodical approach did not present a fully novel method. We modified the text accordingly.

**9. Given the subtitle 'community structure', I had somehow hoped to read a species or genus-level analysis of consumer communities, yet in the end, rotifers were just treated as rotifers, and tardigrades were largely left without a name and assigned to feeding groups, which are well known to be of very limited relevance to describe the actual feeding behaviours of tardigrades, particularly with respect to their potential to feed on microalgae.**

We replaced the community structure by the cryoconite holes community composition and changed the text accordingly.

We are also aware that community structure description lacks detailed information about species composition. Regarding rotifers, the identification depends on the visibility of their coronal cilia which only actively filtrating individuals display. We did not observe a sufficient number of filtrating rotifers during the collection of rotifers for analyses and therefore other identification except few individuals identified as *Macrotrachella* sp. and *Adineta* sp. was impossible.

Regarding the species composition of tardigrades, the representative number of individuals was collected, mounted, and determined apart from individuals intended for analyses. We found *Pilatobius* sp., *Hypsibius* sp., *Hypsibius* cf. *dujardini*, *Isohypsibius* sp. and very rarely *Cryoconicus kaczmareki*. *Pilatobius* sp. was in the past identified as *P. recamieri* which inhabit Arctic tundra commonly. However, utilization of DNA barcoding revealed that suggested *P. recamieri* is a new species for science and it is not formally named yet. *Hypsibius* contains, according to DNA, few cryptic lines. *Grevenius* (previously identified as *Isohypsibius* sp.) is also new for science and not formally named. Moreover, because some species are cryptic and identified only based on DNA (morphologically similar), they most probably use the same food source. Due to the lack of information about the diet of particular species living in cryoconite holes, we chose the trophic group division based on already published knowledge about feeding behaviours and feeding apparatus morphology of related species and on the personal communication with specialists culturing relative species.

**10. So, there is no info on community composition, only some more anecdotal statements as to certain abundantly present Cyanobacteria and algae. This is probably also why there is no statistical analysis of differences in community composition of either consumers or primary producers.**

The data presenting the community structure of primary producers on each glacier were not incorporated into the manuscript because the preservation of the samples by freezing has usually a taxon-specific effect on the survival of the cells of phototrophs. Thus, it makes it impossible to provide reliable quantification comparable between taxa. For example, the cells of glacial algae withstand the repeated freeze-thaw cycles only with a low success, whereas cyanobacteria usually have no significant loss in observed cell numbers. Even though we agree that the information lacks

further data, we felt that it is important to mention at least all information we were able to get from our samples. Due to these reasons, we excluded the data on primary producers from statistical analyses.

The community composition of consumers could not be statistically analysed since were not able to provide sufficient data about the species composition of rotifers among glaciers.

**11. Incidentally, I wondered why both a Spearman rank and a Pearson product-moment correlation were tested. I would expect that if the data fit the assumptions for parametric tests, one would choose the latter, and if not, the former.**

We had to use both tests because some data did not have a normal distribution. However, we were not explicit in the methods that the tests have not been used both on the same data. We modified the text to make the meaning clear.

**12. I would have preferred to see absolute abundances of tardigrades and rotifers in table 1 instead of a general dominance-indication. That dominance indication is not very scientific, and it does not provide any relevant info on potential differences in abundance of consumer groups among gletsjer systems. Actually, since you sampled sufficient replicates in 2 out of 3 gletsjers, you could statistically underpin whether tardigrades were more or less abundant than rotifers in a specific gletsjer system.**

We have changed the table to present frequency of tardigrades and rotifers on each glacier related to the total amount of collected individuals.

**13. Results (Fig. 2): Figure only shows the correlation between cryoconite delta13C and rotifer delta13C. That of tardigrades is not shown because it was not statistically significant. I would then at least like to see the r and actual p-value for the tardigrade correlation, as well as an estimate of the slope of the regression for both rotifers and tardigrades.**

The r in the correlation between $\delta^{13}C$ of decarbonized cryoconite and $\delta^{13}C$ of tardigrades is 0.67 and the *p*-value is 0.07. We added the table and the value to the manuscript.

**14. I would have started the results section with what is now 3.2. Cryoconite composition.**

We transferred the section 3.2 to the beginning of the section Results.

**15. Section 3.3: as mentioned above, table 1 should provide absolute abundances of consumers, and it should also provide info on the proportion of dormant consumers!**

The proportion of dormant consumers is not possible to count during the sample preparation procedure because we use melted material and dormant tardigrades and rotifers are waking up during the whole time of the collecting for isotopic analyses.

**16. I would be interested to read in one or two sentences to what extent other meiofauna or small invertebrates than tardigrades and rotifers were present (any Nematoda, Copepoda, Ostracoda, . . .?).**

Based on the current knowledge, Arctic cryoconite holes are inhabited exclusively by tardigrades and rotifers (e.g. Zawierucha et al., 2018; Zawierucha et al., 2019b). Currently, we have data from 20 glaciers from Svalbard which indicate that no other animals inhabit cryoconite holes in this region (Zawierucha et al. 2020, in review). For our investigation, we used material from many seasons which was analysed by students and experienced researchers and no other metazoans have been found. Moreover, only a few groups of other animals inhabit cryoconite holes worldwide and most

of these species are endemic. Crustaceans (Copepoda) were found only in Himalaya, insects (Chironomidae and Plecoptera) have been found in the Himalaya and Patagonia, recently mites have been found on one Antarctic glacier. Our unpublished meta-analysis suggests that tardigrades and rotifers are the most common cryoconite animals. We corrected the introduction and underlined that exclusively rotifers and tardigrades play roles as apex consumer of cryoconite holes in the Arctic.

**17. At the end of the first discussion §, you finally provide some 'expectation', but it is not clear what this expectation is based upon. In the relatively lengthy and speculative discussion, I read virtually nothing about temperature effects on isotopic fractionation between consumers and resources.**

The temperature in cryoconite holes is very stable (around 0 °C) and the fluctuation usually do not exceed tenths of °C (e.g. Säwström et al., 2002; Zawierucha et al., 2019a). Any increase in the temperature of the sediment is efficiently buffered by ice at 0 °C because any added heat will lead to melting instead. Publications focusing on the temperature effect on the isotopic fractionation usually describe that the changes in fractionation begin with differences higher than 2 °C (e.g. Bosley et al., 2002; Degens et al., 1968; Hinga et al., 1994; Olive et al., 2003).

We added the information about the temperature effect into the discussion and modified the discussion to make its meaning clearer and more explicit.

**18. In section 4.2., dissolved organic matter suddenly becomes an important candidate food for these consumers. That may well be, but what results is this based upon here?**

This suggestion is based on the description and explanation of observed distribution of $\delta^{13}$C in consumers and DOC from cyanobacterial exudates in the study of Velázquez et al. (2017). This study focused, apart from microbial trophic interaction, on isotopic composition of tardigrades and rotifers from Antarctica as well.

We added the reference into the text.

**19. 1) Line 71: "preferential excretion of δ14N" should be "preferential excretion of (light) 14N". 2) Line 250: "rotifers revealed higher values of 15N isotope" should be "rotifers had higher δ15N" or "rotifers had heavier stable nitrogen isotopic signatures'. 3) Line 251: "potential differences in δ15N composition" should be "potential differences in N isotope ratios/signatures".**

We incorporated all above suggested changes and corrections.

**Responses to Mr. Rozwalak**

**1. I would like to get your attention to more precisely distuinguish occurences of primary producers on different glaciers. Currently, it is not clear whether e.g. Ancylonema sp. exist on the all glaciers or only in the Ebbabreen or Svenbreen.**

The data presenting the community structure of primary producers on each glacier were not incorporated into the manuscript because the preservation of the samples by freezing has usually a taxon-specific effect on the survival of the cells of phototrophs. Thus, it makes it impossible to provide quantification. For example, the cells of glacial algae withstand the repeated freeze-thaw cycles only with a low success, whereas cyanobacteria usually have no significant loss in observed cell numbers.

Even though we do have data about the presence or absence of particular species of primary producers on each glacier, we were not able to make a reliable quantitative image of the community structure of primary producers which could help us to elucidate the differences in the stable isotopic composition of cryoconite and invertebrates among glaciers and the differences in trophic groups of tardigrades among glaciers.

Regarding the presence/absence data, all determined species were present on all glaciers except for *Leptolyngbya* sp., which occurred only on Svenbreen and Ebbabreen.

**Responses to Mr. Buda**

**1. You should add a short description or reference how you assessed trophic groups of tardigrades. As a pioneer study, this information may be helpful for future research.**

The description of used methodical approach during the division of tardigrades into trophic groups together with the references is described in methods (part 2.5). We considered using of the term dominant feeding groups and trophic groups as a comparable in case of our study. However, due to the improvement of the comprehensibility of the text, we unified these terms.

**2. Based on visual observation of boxplots in groups, variance looks unequal between groups, it may or may not indicate heteroscedasticity. Variance in Svenn seems to be completely different in comparison to the rest of glaciers. In very low sample size it can highly bias result so it can cause Type I error using ANOVA.**

We are aware that due to the limited number of analysed samples, the visualization of results by boxplots is favouring a bias result. However, we considered boxplots with the visible values as a convenient and clear presentation of our results. Since our work presents a pioneer study, we took into consideration variation in results in the discussion and highlighted the need of further investigation. We mentioned the risk of used visualization in the legend of the Figure 3.

**3. You should consider changing this value to a precise number p and r. Value 0.05 as a threshold is arbitrary, thus it can be important in interpreting results, it can be 0.90 or 0.06.**

The r in the correlation between $\delta^{13}C$ of decarbonized cryoconite and $\delta^{13}C$ of tardigrades is 0.67. We added this value into the results.

**List of all relevant changes made in the manuscript**

**Abstract**

We modified the abstract in order to make it better accessible for general scientific audience and to introduce the modified version of our manuscript properly.

**Introduction**

We added more information about the supraglacial system functioning and the role of cryoconite holes on glaciers. We also added goals and questions of our study.

**Methods**

We added information about the modification of used methods, we also modified the section 2.5 and added information about the taxon-specific effect on the survival of cells of the phototrophs. We added information about correlation coefficients which were used in section 2.6.

**Results**

We transferred the section 3.2 to the beginning of the section results. We also modified Table 1 and added the frequency of consumers on each glacier. The *p*-value of the correlation between $\delta^{13}C$ values of decarbonized cryoconite and the $\delta^{13}C$ of tardigrades was added together with the figure. We changed the name of the section 3.3 and we added more information about the cryoconite community composition.

**Discussion**

The whole discussion was modified in order to put the results better into context and avoid the overinterpretation of the results.

**Conclusions**

Conclusions were slightly modified to summarize the revised manuscript properly.

[revised manuscript text omitted]

---

## Referee Report (RR1)

[referee-annotated manuscript omitted]

---

## Author Response (AR2)

Dear prof. Middelburg, dear Referee,

we are submitting a revised manuscript. Thank you very much for all additional comments and revisions which helped to improve the quality of our manuscript again. We corrected the English as suggested and made some minor changes in the text accordingly to the comments to make its clarity better.

Here, we present the information about substantive changes made in the manuscript.

Yours sincerely,

Tereza Novotná Jaroměřská

**Regarding the two highlighted comments from the Referee:**

*Line125: need to explain WHY this comparison is important in the context of the rest of the intro.*

Based on the current knowledge, we assume that cryoconite consumers are an important component of the nutrient recycling on the glacier surface with the possible impact to ecological processes in downstream areas.

We added this information to the introduction and modified the end of the introduction accordingly to the comments of the Referee.

*Sec 4.1: This paper finds some interesting patterns, but is not able to clearly delineate causes. The authors need to be very clear about the speculative nature of the discussion. I think it's ok, but a statement saying this might be helpful.*

The discussion was modified in parts where our speculations were not clearly separated.

**List of all relevant changes made in the manuscript**

*Introduction*

We modified the end of the discussion accordingly to the comments and improved the consistency of carbon and nitrogen stable isotopes description.

*Results*

We removed the regression line from Fig. 2b because there is no correlation. We also specified the nomenclature of tardigrades according to the newly published study of Zawierucha et al. (2020).

*Discussion*

We modified the text to clearly distinguish our speculations.

**References**

Zawierucha, K., Buda, J., Novotna Jaromerska, T., Janko, K., and Gąsiorek, P.: Integrative approach reveals new species of water bears (*Pilatobius*, *Grevenius*, and *Acutuncus*) from Arctic cryoconite holes, with the discovery of hidden lineages of *Hypsibius*, Zoologischer Anzeiger, 289, 141–165, doi: 10.1016/j.jcz.2020.09.004, 2020.

---

## Author Response (AR3)

Dear prof. Middelburg,

we are submitting a corrected manuscript. Thank you very much for additional comments and corrections. We included all the suggested changes into the text as presented below.

Yours sincerely,

Tereza Novotná Jaroměřská

*l. 32, to increase their surface area (because it refers to ablation zones).*

Corrected as suggested.

*l. 92/93: … favour lighter or discriminate heavier isotopes..*

Corrected as suggested.

*l. 108: unclear sentence: … increased or decreased the uptake of isotopes to keep its isotopic signature… (I do not understand this message)*

This sentence refers to an isotopic homeostasis in organisms. Current version is: "Another study demonstrated, that if the diet is limited by a nutrient, the consumers' body tends to increase or decrease the fractionation against heavier isotope to keep its isotopic values almost constant (Aberle and Malzahn, 2007)".

*l. 127: …emply stable isotope..*

Corrected to: "Here we apply the stable isotope analysis to examine whether the top consumers – tardigrades and rotifers – show probable differences in their food sources in the glacial ecosystem and discuss their trophic position in cryoconite holes."

*l. 182: microL or milliL; do you really add 100 mL, that does not fit in cup!*

Corrected to µl.

*l. 294-296: Have a look at the significance: 53.33% or will 53 or 53.3% do?*

Shortened to integers.

*l. 298: … is not equal among glaciers. If I look at the figure I would write '… is rather similar…'*

Changed to: "slightly different".

*l. 323: Replace on the other hand with however, if you do not use on the one hand as well*

Changed as suggested.

*l. 339: which could be because they potentially consume algae.*

Changed as suggested.

*l. 375-380: write your isotope text simpler. For instance:…. Which would cause depletion in 13C in isotopic signature… Why not: which would lower d13C (values)*

I tried to simplify the text to: "As presented by Post (2002), who focused on freshwater food webs, larger studied lakes evinced higher $\delta^{13}$C values than small lakes suggesting higher occurrence of autochthonous carbon input increasing $\delta^{13}$C of the food web. Based on these findings, we assume that due to its smaller size, Svenbreen may have a higher allochthonous input of nutrients in the form of organic matter from adjacent habitats, which could lower the $\delta^{13}$C because of a longer chain of fractionations discriminating heavier $^{13}$C as it is typical for allochthonous source of carbon (Peterson and Fry, 1987; Post, 2002). Consequently, the depletion in $^{13}$C of consumers on Svenbreen could signify preferential consumption of DOM from the primary production or detritus (Abelson and Hoering, 1961; Iakovenko et al., 2015; Macko and Estep, 1984)."

We also made some corrections through the whole text:

1. isotopic signatures changed to values (as suggested);
2. Figure 3 + Figure 4: correction in the name of Nordenskiöldbreen;
3. small corrections in references.